behaviour/ecology

managed pollinators, pollination ecology, pollen metabarcoding, introduced species, host specialization, diet breadth

**Author for correspondence:**
Anthony D. Vaudo
e-mail: advaudo@gmail.com

# Introduced bees (*Osmia cornifrons*) collect pollen from both coevolved and novel host-plant species within their family-level phylogenetic preferences

Anthony D. Vaudo[1,2], David J. Biddinger[3],
Wiebke Sickel[4,5], Alexander Keller[6]
and Margarita M. López-Uribe[2]

[1]Department of Biology, University of Nevada Reno, Reno, NV 89557, USA
[2]Department of Entomology, Center for Pollinator Research, The Pennsylvania State University, University Park, PA 16802, USA
[3]Fruit Research and Extension Center, The Pennsylvania State University, Biglerville, PA 17307, USA
[4]Thünen Institute of Biodiversity, Johann Heinrich von Thünen Institute, Braunschweig 38116, Germany
[5]Department of Animal Ecology and Tropical Biology, University of Würzburg, Würzburg 97074, Germany
[6]Department of Bioinformatics, University of Würzburg, Center for Computational and Theoretical Biology, Würzburg 97074, Germany

 ADV, 0000-0001-5268-5580; WS, 0000-0002-0038-1478;
AK, 0000-0001-5716-3634; MML-U, 0000-0002-8185-2904

Studying the pollen preferences of introduced bees allows us to investigate how species use host-plants when establishing in new environments. *Osmia cornifrons* is a solitary bee introduced into North America from East Asia for pollination of Rosaceae crops such as apples and cherries. We investigated whether *O. cornifrons* (i) more frequently collected pollen from host-plant species they coevolved with from their geographic origin, or (ii) prefer host-plant species of specific plant taxa independent of origin. To address this question, using pollen metabarcoding, we examined the identity and relative abundance of pollen in larval provisions from nests located in different landscapes with varying abundance of East-Asian and non-Asian plant species. Our results show that *O. cornifrons* collected more pollen from plant species from their native range. Plants in the family Rosaceae were their most preferred pollen hosts, but they

differentially collected species native to East Asia, Europe, or North America depending on the landscape. Our results suggest that while *O. cornifrons* frequently collect pollen of East-Asian origin, the collection of pollen from novel species within their phylogenetic familial affinities is common and can facilitate pollinator establishment. This phylogenetic preference highlights the effectiveness of *O. cornifrons* as crop pollinators of a variety of Rosaceae crops from different geographic origins. Our results imply that globalization of non-native plant species may ease the naturalization of their coevolved pollinators outside of their native range.

## 1. Introduction

The majority of wild and crop plant species require insect-mediated pollination for reproduction [1]. An ideal pollinator comprises particular traits related to its host-plant species: timing its foraging period with blooming period, exhibiting floral constancy to guarantee pollen transfer between conspecific flowers and displaying fidelity to the host-plant species over generations [2,3]. Relationships between pollinators and their host-plant species in their native geographic ranges have evolved over thousands to millions of years, leading to coevolved (morphological, behavioural, chemical and physiological) traits [3,4]. Likewise, efficient crop pollination may result from historical relationships between wild relatives of crops and their native pollinating species [5].

Most crops are cultivated globally, far outside the home range of their wild ancestors and coevolved pollinators [6]. For many pollinator-dependent crops, generalist social bees, such as western honeybees (*Apis mellifera*), are managed to provide crop pollination services. Yet honeybees are not efficient pollinators for many crops (e.g. due to mechanical mismatches of the bee and flower reproductive parts, or lack of sufficient movement between conspecific plants; [7–9]). A logical action to maximize pollination services in agroecosystems is to introduce coevolved crop pollinators into new geographic regions where the crop is cultivated [10–13]. However, the introduction of pollinator species into new habitats poses risks such as competition with native pollinators, pollination of invasive plant species (other than the intended crop) and importation of novel pathogens [14–18].

Several intentionally introduced bees have naturalized outside of their native geographic ranges within short periods of time [14,15,17]. Characterizing the pollen preferences of pollinators in their introduced ranges allows us to test the hypothesis that bees maintain preferences for plant species that they coevolved with in their native range (native preference hypothesis). Alternatively, bees' host-plant preferences could be exclusively driven by phylogenetic affinity independent of the geographic origin of the host-plant (phylogenetic preference hypothesis). These two hypotheses are non-mutually exclusive because bees could prefer host-plants from their native range while keeping their phylogenetic affinity for a limited number of plant lineages. However, support for the phylogenetic hypothesis (i.e. equal use of native and non-native plants from the same plant lineages) would indicate that pollinators could more easily naturalize in new habitats by the utilization of novel, but phylogenetically related, host-plant species [2,14].

In this study, we tested the native and phylogenetic hypotheses with the Japanese orchard bee, *Osmia cornifrons* Radoszkowski 1887 (Hymenoptera: Megachilidae). This solitary univoltine bee was introduced to the United States from East Asia (Japan, Korea, China, Eastern Russia) in 1977 to improve pollination services of fruit trees in the Rosaceae family (e.g. apples, cherries, peaches, pears, etc.; [12,19]) and has supplemented or replaced managed honeybees for fruit tree pollination in North America [20,21]. Phenologically, *O. cornifrons* forage in spring at the onset of blooming of commercial fruit trees in the Rosaceae family, including cherry (native to Europe) followed by peach and apple (native to East Asia). It has been demonstrated that *O. cornifrons* populations in Japan, Russia and the United States exhibit pollen fidelity to Rosaceae and Fabaceae species [22–24], indicating mesolectic foraging behaviour (collecting pollen from many species within one to three main plant families [2]). However, these previous studies analysed pollen via microscopy which only indicated pollen family or genus. Therefore, data are absent regarding the species-level identity of the pollen preferences of *O. cornifrons*, and the degree of pollen collected from native and introduced host-plant species in different landscapes [22–25].

The establishment of *O. cornifrons*, and abundance of crop and wild plant species from East Asia in North America [26,27], makes this an ideal system to determine how floral preferences of recently introduced pollinators may shift in new environments. Using pollen metabarcoding, we analysed pollen species identity and relative abundance in *O. cornifrons* larval pollen provisions. We compared *O. cornifrons* pollen provisions between heterogeneous landscapes of exotic and native plants [28]:

orchards with a high abundance of East-Asian crops (apples and pears), orchards with a high abundance of European crops (cherries) and landscapes with abundance of host-plants from different geographic origins (suburban and wooded forest). We discuss our results in the context of how foraging plasticity to different species and abundant resources within host-plant family preferences can facilitate pollinator establishment in novel habitats.

# 2. Methods

## 2.1. Study system

We placed *O. cornifrons* nests (five boxes of 56-hole BinderBoard® lined with 8 mm diameter × 15 cm paper straws; www.pollinatorparadise.com) on field edges of three predominantly apple and three predominantly cherry orchards, representing typical agricultural landscapes where *O. cornifrons* is managed. We also placed nests in a suburban site (houses within 500 m) and in a wooded forest site (more than 3 km from orchards) representing non-orchard heterogeneous landscapes that *O. cornifrons* occupies (electronic supplementary material, figure S1). All sites were at least 1.8 km away from each other, beyond their reported foraging range [25,29–31]. All females were allowed to forage for their entire active period (approximately four weeks) where one to two cells are provisioned per day in ideal temperature and precipitation conditions [21]. At the end of the growing season, we collected 12 complete larval provisions from cells of different individual nests per site ($N = 96$) where eggs failed to hatch (and were not parasitized by mites [32]). These pollen provisions were selected because they were unused and complete and therefore constitute all pollen provisioned for that cell. No permissions were required prior to conducting field work; all sites are privately owned and work in coordination with Penn State Fruit Research and Extension Center.

## 2.2. Pollen metabarcoding

We extracted DNA using the Qiagen DNeasy Plant Kit. Samples were prepared for sequencing using the pollen metabarcoding protocol for the ITS2 gene region [33,34]. Negative controls were used in PCR to verify there was no plant DNA contamination. Forward and reverse primers contained barcoding sequences so each sample had a unique combination. The final pooled library was spiked with 5% PhiX control to increase sequencing quality. The library was sequenced on an Illumina MiSeq v. 2, 2 × 250 bp spiked with custom index, Read1 and Read2 sequencing primers to bind to the unique ITS2 primers [34].

We filtered forward reads for quality (maxEE = 1) and length (greater than 150 bp). We used the BCdatabaser [35] to subset the NCBI database for the marker and create a reference sequence list of all plant species present in Pennsylvania (https://plants.sc.egov.usda.gov/java/). We classified reads by selecting the match with highest identity to a reference species over the entire amplified region with a global alignment using VSEARCH [36] and a threshold of at least 97% sequence identity. Across all 96 samples, we obtained a total of 6 800 673 reads passing our quality filters, classified 4 922 865 reads and averaged 51 279.84 ± 4872.85 s.e. classified reads per sample. Using the R package phyloseq [37], we imported the final read abundance and taxonomic tables, normalized the read abundance by sample such that each sample totalled 100 (relative read abundances) and filtered taxa that represented less than 1% per sample. We verified the consistency of species identification and abundance by comparing our results to amplicon sequence variants and therefore kept our original dataset for analysis. For classification and filtering scripts, refer to our Dryad data entry (https://doi.org/10.5061/dryad.ffbg79cqn). We also verified consistency in species identification by comparing against the species characterized in the same apple orchards in Kammerer *et al.* [28].

We combined the final abundance and taxonomic tables in JMP Pro 14.3.0 (SAS Institute Inc., Cary, NC, USA). We classified plant species as East-Asian origin, or North American or European. Final filtering removed plant genera that were extremely rare in the dataset, those that contained less than 0.3% of the total counts per site across all sites. Because relative pollen abundances (RRA) by light microscopy and ITS2 metabarcoding have been previously strongly correlated with the primers applied here [33], we used RRA as a proxy for pollen abundance in larval provisions [38–40]. We analysed RRAs by site or landscape using the mean of RRAs from 12 samples per site and used RRAs as a population wide estimate of pollen use and not sample-wise abundance estimations

[38,41,42] (see electronic supplementary material, table S1 for species list and metadata, and for raw data see https://doi.org/10.5061/dryad.ffbg79cqn).

## 2.3. Statistical analysis

To determine diversity of *O. cornifrons* pollen preferences, we estimated and compared Richness, Shannon diversity and Simpson dominance indices between landscapes and sites nested within landscape using generalized linear model (GLM) in JMP (Richness and Simpson with Poisson and Shannon with normal distribution). We used non-metric multidimensional scaling (NMDS) using PRIMER v6 (Plymouth Routines in Multivariate Ecological Research, Auckland, NZ) to determine differences in provisions between landscapes for pollen species or genera. NMDS was conducted using a Bray–Curtis similarity matrix; ANOSIM was conducted with 10 000 permutations to analyse pairwise differences between landscapes.

To test the native hypothesis, we investigated relative proportions of East-Asian versus non-Asian pollen of all plant species across all provisions between landscapes (GLM). To test the phylogenetic hypothesis, we analysed differences in pollen family and genera collected across landscapes (GLM). To test the interaction of the native and phylogenetic hypotheses, we used a $2 \times 2$ chi-square contingency analysis to determine differences in proportions of pollen collected from Asian versus non-Asian, and Rosaceae/Fabaceae versus non-Rosaceae/Fabaceae pollen. These tests were conducted by summing data at the genus level to avoid negative bias, where the average genus representation with high counts is reduced by rare species (with low counts) within the genus. Because *O. cornifrons* exhibited strong preference for Rosaceae, we further investigated differences in the abundance of pollen collected from Asian versus non-Asian Rosaceae species between landscapes with NMDS and GLM. All GLM analyses were conducted in JMP with landscape and site nested in landscape as independent factors, with average RRA as the dependent variable with Poisson distributions; Wilcoxon tests were used for post-hoc comparisons. To reduce repeating significance values, all reported results are significant at $p < 0.05$; $\chi^2$ values are reported for contingency analysis unless GLM is indicated.

## 3. Results

*Osmia cornifrons* pollen provisions comprised five to eight species, one to six genera, and four families on average. Simpson's dominance averaged 0.45 at the genus level indicating low diversity and that this species is indeed a mesolectic forager [2] (i.e. as their collections were dominated by plant species within few plant families [2,28]). Diversity and Simpson's dominance did not differ between landscapes, except species richness was highest in cherry orchards (electronic supplementary material, figure S2; Species Richness). The identity and composition of pollen provisions differed between landscapes, except between apple and suburban sites at the genus level, revealing that suites of pollen collected were landscape dependent (electronic supplementary material, figure S3a,b, species: $R = 0.35$; genus: $R = 0.29$).

Only 18 of 73 plant species we detected with metabarcoding were of East-Asian origin (electronic supplementary material, table S1). Across all samples, the average abundance of East-Asian pollen per sample was higher than non-Asian plants (GLM: $\chi^2 = 840.1$). But this effect was landscape dependent (GLM: $\chi^2 = 1154$): East-Asian pollen was more abundant than non-Asian pollen in apple ($\chi^2 = 1425$) and wooded landscapes ($\chi^2 = 197$), and lower in cherry ($\chi^2 = 37$) and suburban landscapes ($\chi^2 = 93$) (figure 1; electronic supplementary material, figure S4). Rosaceae pollen was by far the most represented among samples (approx. 60% per sample, GLM: $\chi^2 = 24\,700$; electronic supplementary material, figure S5). This was consistent across all sites and landscapes except the suburban site where *Cercis* (Fabaceae) pollen was frequently collected, probably due to the high density of ornamental planting of *C. canadensis* in the urban areas compared to agricultural and wild landscapes (figure 1; electronic supplementary material, figure S4).

We further explored whether *O. corniforns* more frequently collected pollen from East-Asian Rosaceae species versus North American or European Rosaceae species. Indeed, there were differences between landscapes in composition of Rosaceae species collected (figure 2; electronic supplementary material, figure S3c,d, NMDS species: $R = 0.31$; genus: $R = 0.25$) and differences in Asian versus non-Asian Rosaceae pollen collected ($\chi^2 = 851$). East-Asian Rosaceae pollen was more abundant in provisions

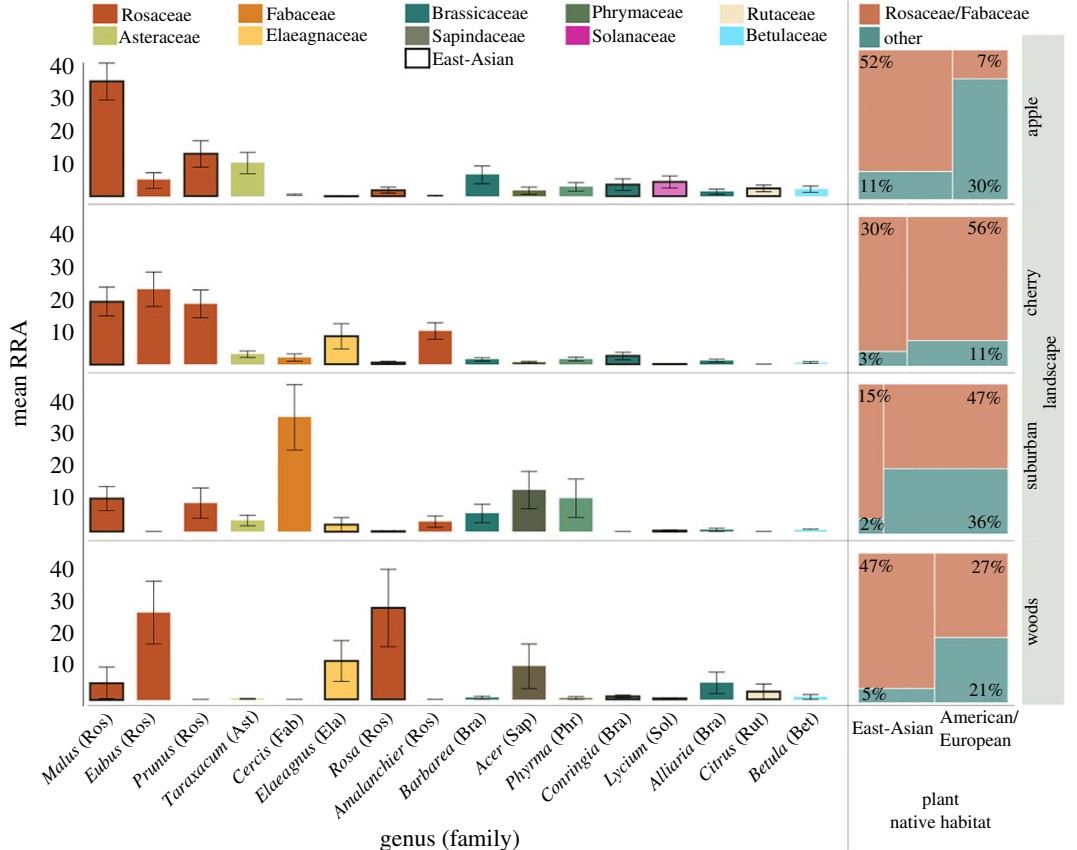

**Figure 1.** *Osmia cornifrons* larval provision host-plant genera by landscape coloured by plant family. Data in left panel are mean relative read abundances ± s.e. Boxes outlined in black are East-Asian origin. Right panel indicates proportions of pollen collected from East-Asian and/or Rosaceae/Fabaceae pollen versus not. Note that Rosaceae genera are the most represented across all landscapes and East-Asian abundances vary by landscape.

than non-Asian Rosaceae pollen in apple orchards ($\chi^2 = 1425$), suburban ($\chi^2 = 280$) and wooded landscapes ($\chi^2 = 197$), and less abundant in cherry orchards ($\chi^2 = 37$; figures 1 and 2).

## 4. Discussion

Our results reveal that *O. cornifrons* collect significant amounts of pollen from Rosaceae host-plants from their native East-Asian geographic origin even in naturalized habitats where alternative host-plants are abundant. However, across sites, they broadly use pollen from Rosaceae species from outside their native range indicating their capability to use host-plants species they did not coevolve with (figures 1 and 2). Our results also identify important secondary pollen hosts (e.g. North American *Cercis canadensis* related to the East-Asian host *C. chinensis*). Overall, *O. cornifrons* foraging is mostly driven by phylogenetic pollen preferences [2,14] showing high representation of rosaceous pollen (detecting Rosaceae species in 96% of all samples) of East-Asian, European and North American origin (electronic supplementary material, figure S4 and S5). The ability of *O. cornifrons* to use pollen sources from related preferred taxa, including crops, may facilitate adaptation to novel environments (figures 1 and 2).

The northeastern United States, where this study occurred, shares the same temperate forest vegetation biome as East Asia (and Northern Europe) and is now home to many East-Asian exotic plant species [26,27]. The pollen provisions of *O. cornifrons* exhibited a heterogeneous mix of East-Asian, European and North American species of *Prunus*, *Rubus* and *Cercis* (genera with native species to each geographic region; figures 1 and 2). Although only 25% of all plant species found in our dataset were introduced to North America from East Asia [26,27], *O. cornifrons* collected significant amounts of pollen from plants from this geographic origin (approx. 40% across landscapes, $\chi^2 = 840.1$, $p < 0.01$; figures 1 and 2; [15]). Interestingly, approximately 90% of the East-Asian pollen collected was in the Rosaceae family (figure 1). As expected, East-Asian pollen from apples (and pears) was dominant in provisions from

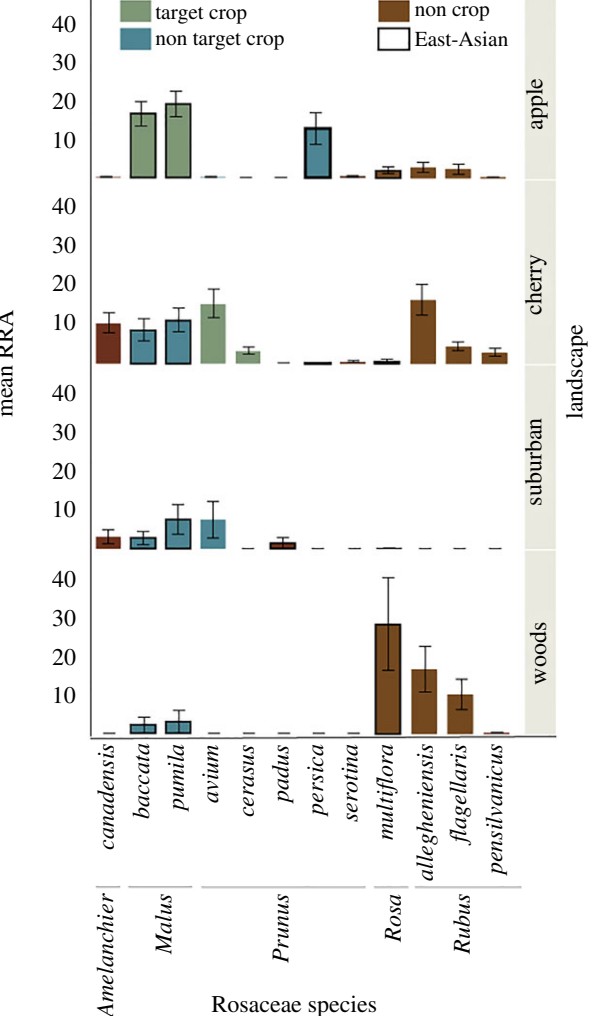

**Figure 2.** Rosaceae plant species in *Osmia cornifrons* larval provisions by landscape. Data are mean relative read abundances ± s.e. Box colour represents target orchard crops of the particular landscape (green), non-target crops (blue), or not crop species (burgundy); boxes outlined in black are East-Asian origin.

apple orchards. However, the nests were placed along field edges where other flowering species and families in edges and forests were available for collection [28] suggesting their intentional choice of crop pollen. In wooded landscapes, where North American native species are more abundant, East-Asian pollen was also over-represented in pollen provisions. In landscapes dominated by European cherry trees, the bees collected pollen from not only cherries, but European, North American, and Asian Rosaceae species evenly, indicating that they foraged on more than just the most abundant host-plant surrounding the nest.

Our results indicate that *O. cornifrons* family-level host-plant preferences drive their foraging behaviour, corroborating that this is a mesolectic species that specializes on a few plant families: Rosaceae and Fabaceae [2]. Given *O. cornifrons*' limited foraging range, the variation between sites that we detected is probably driven by local availability of host-plant species. For instance, where *Cercis* was planted in high abundance in suburban landscapes, as opposed to orchards, we found this type of pollen in high abundance. We did not directly measure flowering plant species abundance at each site during *O. cornifrons*' foraging period, but replication across a variety of sites reveals similar foraging trends. Many plant species from multiple families were blooming at the same time (electronic supplementary material, table S1, figure S4; see [28]), including mass blooming trees such as *Acer* (Sapindaceae) or *Salix* (Salicaceae), which were found in low abundance or not at all in our samples. Additionally, low diversity of larval provisions (composed of multiple foraging trips) indicates *O. cornifrons* forages to a subset of host-plant species available (electronic supplementary material, figure S2 and S4). Our data indicate that East-Asian Rosaceae species are indeed preferred by *O. cornifrons*

[14], but they will use non-Asian Rosaceae host-plants based on local floral availability ([25,28,43]; figures 1 and 2). The use of non-Asian host-plants may be the result of their sampling of the landscape, a by-product of nectar foraging as opposed to active pollen foraging, a dearth of Rosaceae and Fabaceae pollen post-crop bloom in orchards, or that alternative pollen resources are needed in their diet. How bees adjust their foraging behaviour in the absence of preferred resources, or the effects of local floral abundance on species preferences warrants further investigation in more controlled settings (see [22,44]).

The ability of mesolectic bee species to exclusively use host-plant species within family level preferences supports that these pollinators are ideal for crop pollination [17]. In apple orchards, the commercial apple (*Malus pumila*) and its East-Asian pollinizer crabapple (*M. baccata*) were represented evenly in pollen loads, indicating that *O. cornifrons* is effectively delivering pollination services [12], probably visiting both species and transferring pollen within foraging bouts throughout the day. While cherries represent non-Asian host-plant species that may not be their preferred pollen source, it was also actively collected by the bees in cherry orchard landscapes. These results suggest that *O. cornifrons* is an effective pollinator of cherry crops as well. However, we did not observe the same degree of preference for *Prunus* pollen in cherry orchards as we observed for *Malus* pollen in apple orchards (figure 2). In cherry orchards, the bees also visited alternative rosaceous pollen sources (including *Malus*, *Rosa* and *Amelanchier*) which may result from the short cherry blooming phenology or the higher preference for East-Asian pollen from Rosaceae plants. Regardless, our results suggest that the phylogenetic affinity of mesolectic bees for host-plant families that have cosmopolitan distributions can be exploited for agricultural pollination. Another example of this type of relationships can be observed in *Megachile rotundata*, a mesolectic bee specialized on Fabaceae, which is a managed pollinator for a variety of crops such as alfalfa [13].

While mesolectic bees may be ideal candidates for managed crop pollinators, risks associated with the introduction of exotic bee species for pollination services include pollination of invasive plant species and competition with native bee species [14–18]. Although *O. cornifrons* collected pollen from North American native plants (e.g. *Acer*, *Amelanchier*, *Cercis* and *Rubus*), they also collected from invasive species and common weeds of natural and agricultural habitats (e.g. *Barbarea*, *Elaegnus*, *Rosa* and *Taraxacum*) [14]. The quantitative and qualitative results of pollen use that we report in this study suggest that *O. cornifrons* may not be directly competing for floral resources with other native *Osmia* spp. [45,46]. The eastern North American native *O. lignaria lignaria* prefers early-blooming trees such as *Quercus* and *Salix* [45]. Western *O. californica* exhibit preference for *Hydrophyllum*, and *O. l. propinqua* for *Hydrophyllum*, *Salix*, and the widespread *Crataegus* [47,48]. These preferred pollen hosts from North American *Osmia* spp. are absent from our dataset even though they bloom during *O. cornifrons*' foraging period [28]. Potential competitive foraging between *O. cornifrons* and *O. l. lignaria* on *Cercis spp.* could occur as indicated by the preference for *Cercis* pollen in some of our samples [20,22,24,46,49]. However, specific and detailed studies would need to determine the level and effect of competition between species [15,17,46] (including competition for nesting sites in proximity to preferred hosts).

This study provides insights for appropriate management practices of *O. cornifrons*. For instance, managing mesolectic pollinators that prefer pollen from crop plants allows growers to place nests within the field at long distances from edges where pollination services from wild bees are generally reduced [25,29–31,43]. However, bees could be limited to short blooming periods and low nutritional diversity of the crops (such as orchard species), where they still need phenological and phylogenetic diversity of pollen species over their active period (figure 1 for apple orchards; [23,50]). By understanding bees' host-plant preferences, we can recommend complementary native edge and hedgerow host-plant species to support bee populations (e.g. *Rubus* spp. and *C. canadaensis* for *O. cornifrons* in orchards; [22,23,51,52]).

Over 80 bee species have been intentionally or accidentally introduced outside of their native range globally, yet how introduced bees adapt to floral communities outside of their native range is only known for pollen generalist honeybees and bumblebees [15,17]. Our data supports that pollinators show affinity to plants from their native range and will seek these pollen sources in new habitats [14]. This study reveals that despite preferences for a subset of host-plants from their native range as pollen sources, mesolectic bees may successfully establish in new communities when put in contact with novel related host-plants from other geographic origins. Mesolectic foraging, at the plant family level provides a balance between specificity to wild and targeted crop species and the ability to use phylogenetically related pollen sources. This foraging flexibility may be due to phylogenetically conserved floral reward quality [53,54], constraints in bee physiology [55–57], shared phenology between bees and host-plants [43], and bee behaviour and floral morphology [3] that allow bees to respond with plasticity to local resource availability. The globalization of cultivated crop, ornamental and invasive plant species may further facilitate the naturalization of introduced pollinators in novel environments [58].

Data accessibility. Available in Dryad: (i) pollen barcoded sequences, (ii) sample metadata, (iii) ITS2 reference database of plants of Pennsylvania, (iv) code for quality filtering, sequence alignment, and determining taxonomic and OTU tables, (v) resulting combined taxonomic and abundance table, (vi) code for taxa filtering and diversity indices, and (vii) final dataset used for analysis. Supporting figures are available in electronic supplementary material. https://doi.org/10.5061/dryad.ffbg79cqn [59].

Authors' contributions. A.D.V., D.J.B and M.M.L.-U. designed the study. A.D.V. conducted molecular lab work and data analysis. D.J.B. collected pollen samples. W.S. assisted in laboratory techniques. A.K. conducted sequence quality control and classification. A.D.V. and M.M.L.-U. drafted the manuscript. All authors critically revised the manuscript and gave final approval for publication.

Competing interests. We declare we have no competing interests.

Funding. This work was supported by the USDA-SCRI grant PEN04398, USDA-SCRI Coordinated Agricultural Project grant no. MICL05063, USDA Hatch Appropriations under Project PEN04619, and the State Horticultural Association of Pennsylvania.

Permission to carry out fieldwork. No permissions were required prior to conducting field work; all sites are privately owned and work in coordination with Penn State Fruit Research and Extension Center.

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
