## [Reviewer comments · Royal Society Open Science]

Review History

RSOS-200225.R0 (Original submission)

Review form: Reviewer 1

Is the manuscript scientifically sound in its present form?

Yes

Are the interpretations and conclusions justified by the results?

Yes

Is the language acceptable?

Yes

Do you have any ethical concerns with this paper?

No

Have you any concerns about statistical analyses in this paper?

No

Recommendation?

Accept with minor revision (please list in comments)

Comments to the Author(s)

General comments:

In this study, Vaudo et al. examine the foraging preferences of an introduced pollinator, the solitary bee *Osmia cornifrons*, in North American habitats. The hypotheses they propose are that hosts that have co-evolved with *O. cornifrons* will be more attractive than non-native hosts, and that *O. cornifrons* will preferentially forage on species related to their native hosts compared to unrelated hosts. In their study, they placed *O. cornifrons* nests at four location types – apple orchards (one of their native hosts), cherry orchards (same family as native host), a suburban site and a woodland forest. They then used ITS2 metabarcoding on the pollen nest provisions to work out the relative quantities of each pollen type, thus examining the foraging choice of *O. cornifrons* when presented with a range of potential hosts.

The authors have presented a novel and interesting application of pollen metabarcoding, by testing whether there is a native host preference and/or phylogenetic influence on the foraging preferences of *O. cornifrons* in introduced habitats, and how this may influence their ability to adapt outside of their native range. This is a great question that has economically important implications to crop pollination management. For example, can we predict or manage the foraging preferences of introduced pollinators?

The authors found that, in most landscapes, *O. cornifrons* collect significantly more pollen from rosaceous species originating from their native East Asian range. Their study also showed that, when placed at the edge of a rosaceous crop field, *O. cornifrons* is pollinating those crops as anticipated (more so in apple orchards than cherry). As the authors point out, these results provide valuable insights to agricultural managers. The results from the suburban and woodland sites were also very interesting, as mass-flowering crops, I assume, were not within foraging range. In these two sites, they found that *O. cornifrons* will preferentially forage on non-East Asian Fabaceae, but on woodland sites they preferentially forage on both native and non-native Rosaceae. Both of these families have been identified as preferred hosts in their native range, providing support to the hypothesis that the relatedness to native hosts influences the bees' foraging preference.

From the information provided, I wondered if we are seeing an effect of resource availability – bees are merely foraging on what is closest to their nests. It would be good to know more about the composition of vegetation within foraging range, and whether alternative hosts were fully flowering at the time. For example, how close are apple trees to the cherry orchard (and conversely, cherry trees to apple orchards)? *Malus* are clearly within the foraging range of the bee nesting next to cherry orchards, as *Malus* pollen was found in those provisions. Other points of discussion could include: are foraging patterns likely to be different at different times of the year when alternative hosts are flowering? Would placing nests near a crop from a different family (non-rosaceous) yield a dominance of pollen from the unrelated host, or would they still seek out rosaceous hosts?

Another comment is that there are some questions regarding the quantitative potential of metabarcoding, as discussed in the references cited (Deagle et al., Pornon et al., etc), and, more recently, by Bell et al., 2019 and Richardson et al., 2019. A mention of these issues and their possible impact on the interpretation of the results should be included in the discussion. Overall, the authors have explored an important ecological question and have provided good evidence of foraging preferences of *O. cornifrons* under some typical agricultural scenarios using novel molecular methods. I found the manuscript clearly written and enjoyable to read, and it should appeal to a broad audience. I think the manuscript would further benefit from more details about the main concepts being explored (i.e. what we expect under mesolectic vs generalist vs oligolectic foraging preferences) and of the experimental design, as well as a broader discussion about other factors that may be influencing the results observed.

Minor comments:

Line 26. A more detailed explanation of the terms oligolecty and mesolecty and their use in this study would be useful for non-expert readers.

Line 62. How many nests in total were placed at each site?

Line 72. How many nests in total were sampled? How long do the nest provisions take to collect? – this would help us understand over what time period the bees foraged for the pollen provisions.

Line 73. Why are nests collected from nests where eggs failed to hatch (is it because they are intact)? Perhaps clarify this.

Line 77. “spiked with custom index, Read1, and Read2 primers”. Could you clarify what the purpose of this spike-in was? Or should it read 5% PhiX spiked in to the pooled library?

Line 84. Were there any negative controls included that helped inform subsequent filtering?

Line 88. Reference for JMP Pro please.

Line 116. Some statistics on total sequencing reads/reads per sample would be useful. Where there many unidentified/unassigned reads?

Line 117. Does dominance here refer to Simpson dominance index, rather than dominance in the physical sense (e.g. Line 51)? Also please clarify how the results support mesolectic foraging – what would be expected under oligolectic foraging or another scenario?

Line 124. By species analyzed, do you mean detected in metabarcoding?

Line 131. In the urban scenario, how abundant are Fabaceae species (Cercis)?

Line 141. Does "redundancy" here refer to reporting of results? (i.e. to reduce the number of times significance is mentioned?)

Line 147. The authors mention that alternative host plants are abundant; however, it is not clear how this was determined as we don't have information regarding the composition of vegetation at all sites. Were they flowering in similar numbers to the crop plants?

Line 151. “high representation <of> rosaceous pollen (found in 96% of samples)”. Suburban samples had a low representation of rosaceous pollen, but they were few in number, so this comparison is somewhat biased as half of the sites were next to rosaceous crops.

Line 199. Were Crataegus and Salix flowering at the time this study was conducted?

Figure 1. Could you perhaps split Rosaceae and Fabaceae contributions in the right panel to match the left panel?

Figure S2. Consider coloring or separating the different site types to make the figure easier to interpret.

Figure S4. What do mean normalized counts refer to? The number of reads?

Data files. Could the taxonomy database file (its2.penn-malus2018-5.fasta) be included?

Review form: Reviewer 2

Is the manuscript scientifically sound in its present form?

Yes

Are the interpretations and conclusions justified by the results?

No

Is the language acceptable?

Yes

Do you have any ethical concerns with this paper?

No

Have you any concerns about statistical analyses in this paper?

No

Recommendation?

Major revision is needed (please make suggestions in comments)

Comments to the Author(s)

RSOS-200225. Vaudo et al. Phylogenetic pollen preferences facilitate optimal pollination services and may ease the establishment of introduced bees in new habitats

This work examines whether the solitary bee *Osmia cornifrons* shows a pollen-collecting preference for co-evolved plants from its native range and/or a preference for a subset of phylogenetically related plants. The preferences were tested in apple, cherry, woodland and suburban environments.

The manuscript is well laid out, generally clearly written, and brief. This is an interesting topic and I enjoyed reading the manuscript. I've made some comments directly in the pdf (Appendix A).

While I note the aims of the work (per the abstract), overall, I found the hypothesis and conclusions somewhat unclear and slightly frustrating. The 'and/or' nature of the aims perhaps contributes to this. For phylogenetic preference, the authors note that *O. cornifrons* is mesolectic for Rosaceae and Fabaceae (L37/38). Was this in doubt? Was there – for example – any question that *O. cornifrons* would collect cherry pollen, in preference to non Rosaceae/Fabaceae alternatives? If there was room for discovery in this area, it would be useful if this was more specifically outlined.

With less unknowns (I think) with respect to phylogenetic preference, the focus was seemingly on the tricky task of identifying native range preferences in the field. How to separate these preferences from attendance at the dominant available pollen source? It is difficult for me to see how the native range hypothesis would be properly tested unless the bees were placed in proximity to closely related plant species (such as *Malus*?) of different geographical origin. Some quantification of the available pollen resources would be useful and it's unclear to me if the authors tried to do this? It certainly isn't prominently described.

The authors strongly suspect that *O. cornifrons* will gather pollen from the dominant locally available Rosaceae and Fabaceae; "Therefore, we predict that *O. cornifrons* would exhibit foraging preference for East-Asian Rosaceae and Fabaceae in landscapes dominated by these plants (apple orchards and suburban areas). In contrast, we predict that *O. cornifrons* would forage more on European or North American rosaceous and fabaceous species in landscapes where these plants are more abundant (cherry orchards and wooded forests)" (L51-55). But the authors don't really explain how they will get around this problem. Finally, in the last final sentence of the Introduction, the authors seemingly hypothesise against the native range preference.

While these points above are largely about the set-up of the manuscript, they are then, naturally, reflected in the Results and Discussion. There is a lack of firmness and structure about what was tested and what was concluded. Rather, the observations of pollen collected - in very different landscapes - are correlative/indicative (see the title itself).

In a study of native range preferences, taxonomic IDs to species level (at least best matches to species) must be important. Averaged details of the pollen IDs appear in the first line of the results, including a mention of species detected. However, the results for the species-level identifications are very scant. It may be that this information is given in the Dryad archive (which appears to be extensive), but this is not mentioned/cited in the text, and I cannot access the Dryad material. I think the authors really should present a table or diagram that summarises the species-level findings in the text (or supp mat).

Why is it necessary to present Fig 1 at genus level? Is this because the authors are not confident of the ID to species level? Is there any possibility that a genus-level category might contain more than one species? It is noted at L182, pollen from both apple and crabapple were detected in the apple orchards.

What and where were the *Malus* and the *Rubus* sp in the cherry orchard?

I found the methods used for sequence processing and taxonomic identification to be slightly opaque (comments on L79-93). As one small point, the authors state that species identifications were verified, but I'd prefer to know how these were verified. A presentation of the species-level results might help illuminate the methods used.

Decision letter (RSOS-200225.R0)

Dear Dr Vaudo,

The editors assigned to your paper ("Phylogenetic pollen preferences facilitate optimal pollination services and may ease the establishment of introduced bees in new habitats") have now received comments from reviewers. We would like you to revise your paper in accordance with the referee and Associate Editor suggestions which can be found below (not including confidential reports to the Editor). Please note this decision does not guarantee eventual acceptance.

Please submit a copy of your revised paper before 28-Jun-2020. Please note that the revision deadline will expire at 00.00am on this date. If we do not hear from you within this time then it will be assumed that the paper has been withdrawn. In exceptional circumstances, extensions may be possible if agreed with the Editorial Office in advance. We do not allow multiple rounds of revision so we urge you to make every effort to fully address all of the comments at this stage. If deemed necessary by the Editors, your manuscript will be sent back to one or more of the original reviewers for assessment. If the original reviewers are not available, we may invite new reviewers.

- Data accessibility

It is a condition of publication that all supporting data are made available either as supplementary information or preferably in a suitable permanent repository. The data

accessibility section should state where the article's supporting data can be accessed. This section should also include details, where possible of where to access other relevant research materials such as statistical tools, protocols, software etc can be accessed. If the data have been deposited in an external repository this section should list the database, accession number and link to the DOI for all data from the article that have been made publicly available. Data sets that have been deposited in an external repository and have a DOI should also be appropriately cited in the manuscript and included in the reference list.

If you wish to submit your supporting data or code to Dryad (<http://datadryad.org/>), or modify your current submission to dryad, please use the following link:
<http://datadryad.org/submit?journalID=RSOS&manu=RSOS-200225>

- **Competing interests**

- **Authors' contributions**

- **Acknowledgements**

- **Funding statement**

on behalf of Dr Sean Rands (Associate Editor) and Pete Smith (Subject Editor)
openscience@royalsociety.org

Associate Editor's comments (Dr Sean Rands):

Associate Editor: 1

Comments to the Author:

Thank you for submitting your manuscript to Royal Society Open Science. Two referees have now provided their comments on your manuscript.

Please ensure that you address their comments fully while revising your manuscript, and provide a point-by-point response upon resubmission.

Comments to Author:

Reviewers' Comments to Author:

Reviewer: 1

Comments to the Author(s)

General comments:

In this study, Vaudo et al. examine the foraging preferences of an introduced pollinator, the solitary bee *Osmia cornifrons*, in North American habitats. The hypotheses they propose are that hosts that have co-evolved with *O. cornifrons* will be more attractive than non-native hosts, and that *O. cornifrons* will preferentially forage on species related to their native hosts compared to unrelated hosts. In their study, they placed *O. cornifrons* nests at four location types – apple orchards (one of their native hosts), cherry orchards (same family as native host), a suburban site and a woodland forest. They then used ITS2 metabarcoding on the pollen nest provisions to work out the relative quantities of each pollen type, thus examining the foraging choice of *O. cornifrons* when presented with a range of potential hosts.

The authors have presented a novel and interesting application of pollen metabarcoding, by testing whether there is a native host preference and/or phylogenetic influence on the foraging preferences of *O. cornifrons* in introduced habitats, and how this may influence their ability to adapt outside of their native range. This is a great question that has economically important implications to crop pollination management. For example, can we predict or manage the foraging preferences of introduced pollinators?

The authors found that, in most landscapes, *O. cornifrons* collect significantly more pollen from rosaceous species originating from their native East Asian range. Their study also showed that, when placed at the edge of a rosaceous crop field, *O. cornifrons* is pollinating those crops as anticipated (more so in apple orchards than cherry). As the authors point out, these results provide valuable insights to agricultural managers. The results from the suburban and woodland sites were also very interesting, as mass-flowering crops, I assume, were not within foraging range. In these two sites, they found that *O. cornifrons* will preferentially forage on non-East Asian Fabaceae, but on woodland sites they preferentially forage on both native and non-native Rosaceae. Both of these families have been identified as preferred hosts in their native range, providing support to the hypothesis that the relatedness to native hosts influences the bees' foraging preference.

From the information provided, I wondered if we are seeing an effect of resource availability – bees are merely foraging on what is closest to their nests. It would be good to know more about the composition of vegetation within foraging range, and whether alternative hosts were fully flowering at the time. For example, how close are apple trees to the cherry orchard (and conversely, cherry trees to apple orchards)? *Malus* are clearly within the foraging range of the bee nesting next to cherry orchards, as *Malus* pollen was found in those provisions. Other points of discussion could include: are foraging patterns likely to be different at different times of the year when alternative hosts are flowering? Would placing nests near a crop from a different family (non-rosaceous) yield a dominance of pollen from the unrelated host, or would they still seek out rosaceous hosts?

Another comment is that there are some questions regarding the quantitative potential of metabarcoding, as discussed in the references cited (Deagle et al., Pornon et al., etc), and, more recently, by Bell et al., 2019 and Richardson et al., 2019. A mention of these issues and their possible impact on the interpretation of the results should be included in the discussion.

Overall, the authors have explored an important ecological question and have provided good evidence of foraging preferences of *O. cornifrons* under some typical agricultural scenarios using novel molecular methods. I found the manuscript clearly written and enjoyable to read, and it should appeal to a broad audience. I think the manuscript would further benefit from more details about the main concepts being explored (i.e. what we expect under mesolectic vs generalist vs oligolectic foraging preferences) and of the experimental design, as well as a broader discussion about other factors that may be influencing the results observed.

Minor comments:

Line 26. A more detailed explanation of the terms oligolecty and mesolecty and their use in this study would be useful for non-expert readers.

Line 62. How many nests in total were placed at each site?

Line 72. How many nests in total were sampled? How long do the nest provisions take to collect? – this would help us understand over what time period the bees foraged for the pollen provisions.

Line 73. Why are nests collected from nests where eggs failed to hatch (is it because they are intact)? Perhaps clarify this.

Line 77. “spiked with custom index, Read1, and Read2 primers”. Could you clarify what the purpose of this spike-in was? Or should it read 5% PhiX spiked in to the pooled library?

Line 84. Were there any negative controls included that helped inform subsequent filtering?

Line 88. Reference for JMP Pro please.

Line 116. Some statistics on total sequencing reads/reads per sample would be useful. Where there many unidentified/unassigned reads?

Line 117. Does dominance here refer to Simpson dominance index, rather than dominance in the physical sense (e.g. Line 51)? Also please clarify how the results support mesolectic foraging – what would be expected under oligolectic foraging or another scenario?

Line 124. By species analyzed, do you mean detected in metabarcoding?

Line 131. In the urban scenario, how abundant are Fabaceae species (*Cercis*)?

Line 141. Does "redundancy" here refer to reporting of results? (i.e. to reduce the number of times significance is mentioned?)

Line 147. The authors mention that alternative host plants are abundant; however, it is not clear how this was determined as we don't have information regarding the composition of vegetation at all sites. Were they flowering in similar numbers to the crop plants?

Line 151. “high representation rosaceous pollen (found in 96% of samples)”. Suburban samples had a low representation of rosaceous pollen, but they were few in number, so this comparison is somewhat biased as half of the sites were next to rosaceous crops.

Line 199. Were *Crataegus* and *Salix* flowering at the time this study was conducted?

Figure 1. Could you perhaps split Rosaceae and Fabaceae contributions in the right panel to match the left panel?

Figure S2. Consider coloring or separating the different site types to make the figure easier to interpret.

Figure S4. What do mean normalized counts refer to? The number of reads?

Data files. Could the taxonomy database file (its2.penn-malus2018-5.fasta) be included?

Reviewer: 2

Comments to the Author(s)

RSOS-200225. Vaudo et al. Phylogenetic pollen preferences facilitate optimal pollination services and may ease the establishment of introduced bees in new habitats

This work examines whether the solitary bee *Osmia cornifrons* shows a pollen-collecting preference for co-evolved plants from its native range and/or a preference for a subset of phylogenetically related plants. The preferences were tested in apple, cherry, woodland and suburban environments.

The manuscript is well laid out, generally clearly written, and brief. This is an interesting topic and I enjoyed reading the manuscript. I've made some comments directly in the pdf.

While I note the aims of the work (per the abstract), overall, I found the hypothesis and conclusions somewhat unclear and slightly frustrating. The 'and/or' nature of the aims perhaps contributes to this. For phylogenetic preference, the authors note that *O. cornifrons* is mesolectic for Rosaceae and Fabaceae (L37/38). Was this in doubt? Was there – for example - any question that *O. cornifrons* would collect cherry pollen, in preference to non Rosaceae/Fabaceae alternatives? If there was room for discovery in this area, it would be useful if this was more specifically outlined.

With less unknowns (I think) with respect to phylogenetic preference, the focus was seemingly on the tricky task of identifying native range preferences in the field. How to separate these preferences from attendance at the dominant available pollen source? It is difficult for me to see how the native range hypothesis would be properly tested unless the bees were placed in proximity to closely related plant species (such as *Malus*?) of different geographical origin. Some quantification of the available pollen resources would be useful and it's unclear to me if the authors tried to do this? It certainly isn't prominently described.

The authors strongly suspect that *O. cornifrons* will gather pollen from the dominant locally available Rosaceae and Fabaceae; "Therefore, we predict that *O. cornifrons* would exhibit foraging preference for East-Asian Rosaceae and Fabaceae in landscapes dominated by these plants (apple orchards and suburban areas). In contrast, we predict that *O. cornifrons* would forage more on European or North American rosaceous and fabaceous species in landscapes where these plants are more abundant (cherry orchards and wooded forests)" (L51-55). But the authors don't really explain how they will get around this problem. Finally, in the last final sentence of the Introduction, the authors seemingly hypothesise against the native range preference.

While these points above are largely about the set-up of the manuscript, they are then, naturally, reflected in the Results and Discussion. There is a lack of firmness and structure about what was tested and what was concluded. Rather, the observations of pollen collected - in very different landscapes - are correlative/indicative (see the title itself).

In a study of native range preferences, taxonomic IDs to species level (at least best matches to species) must be important. Averaged details of the pollen IDs appear in the first line of the results, including a mention of species detected. However, the results for the species-level identifications are very scant. It may be that this information is given in the Dryad archive (which appears to be extensive), but this is not mentioned/cited in the text, and I cannot access the Dryad material. I think the authors really should present a table or diagram that summarises the species-level findings in the text (or supp mat).

Why is it necessary to present Fig 1 at genus level? Is this because the authors are not confident of the ID to species level? Is there any possibility that a genus-level category might contain more than one species? It is noted at L182, pollen from both apple and crabapple were detected in the apple orchards.

What and where were the *Malus* and the *Rubus* sp in the cherry orchard?

I found the methods used for sequence processing and taxonomic identification to be slightly opaque (comments on L79-93). As one small point, the authors state that species identifications were verified, but I'd prefer to know how these were verified. A presentation of the species-level results might help illuminate the methods used.

Author's Response to Decision Letter for (RSOS-200225.R0)

See Appendix B.

Decision letter (RSOS-200225.R1)

Dear Dr Vaudo,

It is a pleasure to accept your manuscript entitled "Introduced bees (*Osmia cornifrons*) collect pollen from both coevolved and novel host-plant species within their family-level phylogenetic preferences" in its current form for publication in Royal Society Open Science. The comments of the reviewer(s) who reviewed your manuscript are included at the foot of this letter.

on behalf of Dr Sean Rands (Associate Editor) and Pete Smith (Subject Editor)
openscience@royalsociety.org

Associate Editor Comments to Author (Dr Sean Rands):

Comments to the Author:

Thankyou for the detailed response in your revision. I am convinced that you have addressed the reviewers' comments appropriately, and am happy to recommend this for publication.

Appendix A**ROYAL SOCIETY
OPEN SCIENCE****Phylogenetic pollen preferences facilitate optimal
pollination services and may ease the establishment of
introduced bees in new habitats**

Journal:	Royal Society Open Science
Manuscript ID	RSOS-200225
Article Type:	Research
Date Submitted by the Author:	11-Feb-2020
Complete List of Authors:	Vaudo, Anthony; University of Nevada Reno, Department of Biology; Pennsylvania State University University Park, Department of Entomology Biddinger, David; Pennsylvania State University Fruit Research and Extension Center, Entomology Sickel, Wiebke; Thunen Institute for Biodiversity, ; Universität Würzburg, Animal Ecology and Tropical Biology Keller, Alexander; University of Würzburg, Department of Animal Ecology and Tropical Biology Lopez-Uribe, Margarita M; Pennsylvania State University, Entomology
Subject:	behaviour < BIOLOGY, ecology < BIOLOGY
Keywords:	managed pollinators, pollination ecology, pollen metabarcoding, introduced species, host specialization, diet breadth
Subject Category:	Ecology, Conservation, and Global Change Biology

Author-supplied statements

Relevant information will appear here if provided.

Ethics

Does your article include research that required ethical approval or permits?:

This article does not present research with ethical considerations

Statement (if applicable):

CUST_IF_YES_ETHICS :No data available.

Data

It is a condition of publication that data, code and materials supporting your paper are made publicly available. Does your paper present new data?:

Yes

Statement (if applicable):

Available in Dryad: 1) Pollen barcoded sequences, 2) Sample metadata, 3) Code for quality filtering, sequence alignment, and determining taxonomic and OTU tables, 4) Resulting combined taxonomic and abundance table, 5) Code for taxa filtering and diversity indices, and 6) Final dataset used for analysis. Supporting figures are available in Electronic Supplementary Material.

<https://doi.org/10.5061/dryad.ffbg79cqn>

<https://datadryad.org/stash/share/ax980yCG8bx2VbyI9HdmPSqkkYy0PZgdYJhxD59PYhA>

Conflict of interest

I/We declare we have no competing interests

Statement (if applicable):

CUST_STATE_CONFLICT :No data available.

Authors' contributions

This paper has multiple authors and our individual contributions were as below

Statement (if applicable):

ADV, DJB, and MMLU designed the study. ADV conducted molecular lab work and data analysis. DJB collected pollen samples. WS assisted in lab techniques. AK conducted sequence quality control and classification. ADV and MMLU drafted the manuscript. All authors critically revised the manuscript and gave final approval for publication.

Phylogenetic pollen preferences facilitate optimal pollination services and may ease the establishment of introduced bees in new habitats

Anthony D. Vaudo (ORCID: 0000-0001-5268-5580)^{*1,2}, David J. Biddinger³, Wiebke Sickel (ORCID: 0000-0002-0038-1478)^{4,5}, Alexander Keller (ORCID: 0000-0001-5716-3634)⁶, and Margarita M. López-Urbe (ORCID: 0000-0002-8185-2904)²

1. University of Nevada Reno, Department of Biology, Reno, NV, USA 89557

2. The Pennsylvania State University, Department of Entomology, Center for Pollinator Research, University Park, PA, USA 16802

3. The Pennsylvania State University, Fruit Research and Extension Center, Biglerville, PA, USA 17307

4. Thünen Institute, Thünen Institute for Biodiversity, Braunschweig, Germany 38116

5. University of Würzburg, Department of Animal Ecology and Tropical Biology, Würzburg, Germany 97074

6. University of Würzburg, Department of Bioinformatics, Center for Computational and Theoretical Biology, Würzburg, Germany 97074

Abstract

Studying the pollen preferences of bees outside of their native range allows us to investigate how species utilize host-plants when establishing in new environments. We investigated whether *Osmia cornifrons*, a solitary bee introduced into North America from East-Asia for pollination of rosaceous crops, prefer host-plant species from their geographic origin and/or prefer hosts of particular plant taxa, facilitating switching to new host species in their introduced range. We tested these two non-exclusive hypotheses using pollen metabarcoding of larval provisions from nests located in habitats with varying abundance of East-Asian and non-Asian plant species. We found that the bees disproportionately, yet not exclusively, collected pollen from their native range. Plants in the family Rosaceae were their most preferred pollen hosts, where species native to East-Asia, Europe, or North America were differentially collected dependent on landscape. Our results suggest that phylogenetic host-plant preferences can facilitate pollinator establishment by switching to novel host-plant species within their familial affinities. This phylogenetic preference highlights their effectiveness as crop pollinators of a variety of rosaceous crops from different geographic origins. These data imply that the globalization of non-native plant species may ease the naturalization of introduced pollinators in non-native habitats worldwide.

Key Words

managed pollinators, pollination ecology, pollen metabarcoding, introduced species, host specialization, diet breadth

Introduction

The majority of wild and crop plant species require insect mediated pollination for reproduction[1].
An ideal pollinator comprises particular traits related to its host-plant species: timing its foraging
period with flowering phenology, showing floral constancy that guarantees intraspecific pollen
transfer between conspecific flowers, and displaying fidelity to the host-plant species over
generations[2,3]. Relationships between pollinators and their host-plant species in their native
geographic ranges have evolved over thousands to millions of years, leading to coevolved traits[4].
Likewise, efficient crop pollination may result from historical relationships between wild relatives
of crops and their native pollinating species[5].

Most crops are cultivated globally, far outside the home range of their wild ancestors and
coevolved pollinators[6]. For many pollinator-dependent crops, generalist social bees, such as
honey bees (*Apis mellifera*), are managed to provide crop pollination services. Yet honey bees are
not efficient pollinators for many crops because of mechanical or nutritional mismatches[7-9]. A
logical action to maximize pollination services in agroecosystems is to introduce the coevolved
crop pollinator into new geographic regions where the crop is cultivated[10-13]. However, the
introduction of pollinator species into new habitats poses risks such as competition with native
pollinators, pollination of invasive plant species (other than the intended crop), and importation of
novel pathogens[14-18].

Several purposefully introduced bees have naturalized outside of their native geographic ranges
within short periods of time[14,15,17]. Characterizing the pollen preferences of pollinators in their
introduced ranges allows us to test two non-mutually exclusive hypotheses relating to the
establishment of pollinators in new habitats: 1) bees maintain preference for plant species that they
coevolved with in their native range (native preference hypothesis) and 2) bees' host-plant
preferences are driven by phylogenetic affinity (i.e. oligolecty or mesolecty [2]; phylogenetic
preference hypothesis). These dietary constraints could facilitate the utilization of novel, but
phylogenetically related, host-plant species when introduced to new habitats[2,14].

In this study, we tested the native and phylogenetic hypotheses ~~above~~ with the Japanese Orchard
Bee, *Osmia cornifrons* Radoszkowski 1887 (Hymenoptera: Megachilidae). This solitary
univoltine bee was introduced to the United States from East-Asia (Japan, Korea, China, Eastern
Russia) in 1977 to improve pollination services of rosaceous fruit trees[12,19], and has
supplemented or replaced managed honey bees for fruit tree pollination in North America[20,21].
Phenologically, *O. cornifrons* forage in Spring at the onset of blooming of commercial rosaceous
fruit trees; including cherry (native to Europe) followed by peach and apple (native to East-Asia).
It has been demonstrated that *Osmia cornifrons* exhibit pollen fidelity to Rosaceae and Fabaceae
species indicating mesolectic foraging behavior[22-24]. However, data are absent regarding the
degree of pollen foraging from landscapes that vary in the availability of native and introduced
host-plants species[22-25].

The establishment of *O. cornifrons*, and abundance of crop and wild plant species from East-Asia
in North America[26, 27], makes this an ideal system to determine how floral preferences of
recently introduced pollinators may shift in new environments. Here, we analyzed pollen identity
and abundance of *O. cornifrons* larval pollen provisions as a proxy of host-plant preference in
heterogeneous landscapes of exotic and native plants[28]: orchards with a dominance of East-
Asian crops (apples and pears), orchards with a dominance of European crops (cherries), and
landscapes with abundance of host-plants from different geographic origins (suburban and wooded
forest). We predict that *O. cornifrons* would exhibit foraging preference for East-Asian Rosaceae
and Fabaceae, yet landscape trends in geographic origin of rosaceous and fabaceous pollen
collected will reflect their dominance of each landscape. Therefore, we predict that *O. cornifrons*
would exhibit foraging preference for East-Asian Rosaceae and Fabaceae in landscapes dominated
by these plants (apple orchards and suburban areas). In contrast, we predict that *O. cornifrons*
would forage more on European or North American rosaceous and fabaceous species in landscapes
where these plants are more abundant (cherry orchards and wooded forests). We argue that the
plasticity to forage on abundant resources within host-plant phylogenetic preferences, but
independently of the area of origin of the plant, can facilitate pollinator establishment in novel
habitats.

**Methods**

*Study System*

We placed *O. cornifrons* nests (56-hole BinderBoard® lined with 8mm diameter x 15cm paper
straws; www.pollinatorparadise.com) on field edges of three predominantly apple and three
predominantly cherry orchards, representing typical agricultural landscapes where *O. cornifrons*
is managed. We also placed nests in a suburban site (houses within 500m) and in a wooded forest
site (>3km from orchards) representing non-orchard heterogeneous landscapes that *O. cornifrons*
occupies (Figure S1). All sites were ≥ 1.8 km away from each other, beyond their reported foraging
range[25,29-31]. No permissions were required prior to conducting field work; all sites are
privately owned and work in coordination with Penn State Fruit Research and Extension Center.

*Pollen Metabarcoding*

We collected 12 complete pollen provisions from different nests per site (N=96) from cells in
which eggs failed to hatch (and not attacked by mites[32]). We extracted DNA using the Qiagen
DNeasy Plant Kit. Samples were prepared for sequencing using the pollen metabarcoding protocol
for the ITS2 gene region[33,34]. Forward and reverse primers contained barcoding sequences so
each sample had a unique combination. The final pooled library with 5% PhiX was sequenced on
an Illumina MiSeq v2 2x250bp, spiked with custom index, Read1, and Read2 primers.

We filtered forward reads for quality (maxEE=1) and length (>150bp). We used the
Bcdatabaser[35] to subset the NCBI database for the marker and a list of all plant species present
51 52 53 54 55 56 57 58 59 60

in Pennsylvania (<https://plants.sc.egov.usda.gov/java/>). We classified reads to species using
VSEARCH[36] with a threshold of >97% sequence identity. Using the R package phyloseq[37],
we imported the final read abundance and taxonomic tables, and relativized abundance by sample
and filtered taxa that represented <1% (<https://doi.org/10.5061/dryad.ffbg79cqn>). We verified the
accuracy of species identification by comparison to amplicon sequence variants and verified
species presence from the same apple orchards characterized in Kammerer et al. 2016[28].

We combined the final abundance and taxonomic tables in JMP Pro 14.3.0. We classified plant
species of East-Asian origin, or North American or European. Final filtering removed plant genera
that contained less than 0.3% of the total counts per site. Because relative pollen abundances by
light microscopy and ITS2 metabarcoding were previously strongly correlated with the primers
applied here[33], we used relative read abundances as a proxy for pollen abundance in larval
provisions[38-40].

*Statistical Analysis*

To determine diversity of *O. cornifrons* pollen preferences, we estimated Richness, Shannon
diversity, and Simpson dominance indices with GLM (Richness and Simpson with Poisson and
Shannon with normal distribution). We used non-metric multidimensional scaling (NMDS) to
determine differences in provisions between landscapes for pollen species or genera. NMDS was
conducted using a Bray-Curtis similarity matrix; ANOSIM was conducted with 10,000
permutations to analyze pairwise differences between landscapes.

To test the native preference hypothesis, we investigated relative proportions of East-Asian vs.
non-Asian pollen of all plant species across all provisions between landscapes (GLM). To test the
phylogenetic preference hypothesis, we analyzed differences in pollen family and genera collected
across landscapes (GLM). To test the interaction of the native and phylogenetic hypotheses, we
used a 2x2 Chi-square contingency analysis to determine differences in proportions of pollen
collected from Asian vs. non-Asian, and rosaceous/fabaceous vs. non-rosaceous/fabaceous pollen.
Because *O. cornifrons* exhibited strong preference for Rosaceae, we further investigated
differences in the abundance of pollen collected from Asian vs non-Asian Rosaceae species
between landscapes (NDMS; GLM). All GLM analyses were conducted in JMP with landscape,
and site nested in landscape, as independent factors with average count of normalized reads as the
dependent variable with Poisson distributions; Wilcoxon tests were used for post-hoc comparisons.

**Results**

*Osmia cornifrons* pollen provisions comprised 5-8 species, 1-6 genera, and 4 families on average,
and dominance averaged 0.52 supporting mesolectic foraging[2] (i.e. their collections were
dominated by a subset of available plant species and families[28]). Diversity and dominance did
not differ between landscapes, except species richness was highest in cherry orchards (Figure S2;
Species Richness). The composition of pollen provisions differed between landscapes, except

between apple and suburban sites at the genus level, revealing that suites of pollen collected were
landscape dependent (Figure S3ab, species: $R=0.35$; genus: $R=0.29$).

Only 18 of 73 plant species analyzed were of East-Asian origin. Yet, across all samples, the
average abundance of East-Asian pollen per sample was higher than non-Asian plants
(GLM: $X^2=840.1$). But as predicted, this effect was landscape dependent (GLM: $X^2=1154$): East-
Asian pollen was more abundant than non-Asian pollen in apple ($X^2=1425$) and wooded
landscapes ($X^2=197$), and lower in cherry ($X^2=37$) and suburban landscapes ($X^2=93$) (Figure 1).
Rosaceous pollen was by far the most represented among samples (~60% per sample,
GLM: $X^2=24700$; Figure S4). This was consistent across all landscapes except the suburban site
where *Cercis* (Fabaceae) pollen was also frequently collected (Figure 1, Figure S4).

We further explored whether *O. cornifrons* exhibits foraging preferences within Rosaceae for East-
Asian species versus North American or European species. Indeed, there were differences between
landscapes in composition of Rosaceae species collected (Figure S3cd, NMDS species: $R=0.31$;
genus: $R=0.25$) and differences in Asian vs non-Asian Rosaceae pollen collected ($X^2=851$). East-
Asian Rosaceae pollen was more abundant in provisions than non-Asian Rosaceae pollen in apple
orchards ($X^2=1425$), suburban ($X^2=280$) and wooded landscapes ($X^2=197$), and less abundant in
cherry orchards ($X^2=37$; Figure 1,2). This emphasizes that in new habitats, *O. cornifrons*' affinity
to Rosaceae facilitates shifting to novel rosaceous host-plant species, including crops, that
originated outside of their native range. To reduce redundancy, all reported results are significant
at $P<0.05$; X^2 values are reported for contingency analysis unless GLM is indicated.

Discussion

Our results reveal that *O. cornifrons* collect significant amounts of pollen from rosaceous host-
plants from their native East-Asian geographic origin (apple orchards and wooded forest) even in
naturalized habitats where alternative host-plants are abundant. However, across sites, they
broadly utilize pollen from Rosaceae species evolved from outside their native range. Our results
also identify important secondary pollen hosts (e.g. North American *Cercis canadensis* related to
the East-Asian host *C. chinensis*). Overall, *O. cornifrons* foraging is mostly driven by phylogenetic
pollen preferences[2,14] showing high representation rosaceous pollen (found in 96% of samples)
of East-Asian, European and North American origin. Their mesolectic diet preference allows these
bees to utilize pollen sources from related preferred taxa which may facilitate adaptation to novel
environments (Figure 1,2).

The Northeastern USA shares the same temperate forest vegetation biome as East-Asia (and
Northern Europe) and is now home to many East-Asian exotic plant species[26,27]. The pollen
provisions of *O. cornifrons* exhibited a heterogenous mix of East-Asian, European and North
American species of *Prunus*, *Rubus*, and *Cercis* (genera with native species to each geographic
region; Figure 2). Although only ~25% of all plant species found in our datasets were introduced

to North America from East-Asia[26,27], *O. cornifrons* collected significant amounts of pollen
from plants from this geographic origin (40% across landscapes, $X^2=840.1$, $P < 0.01$; Figure
1,2)[15]. Interestingly, approximately 90% of the East-Asian pollen collected was rosaceous
(Figure 1). As expected, East-Asian pollen was more represented in provisions from apple
orchards. However, the nests were placed along field edges where flowering species in edges and
forests were available for collection[28] suggesting their intentional choice of crop pollen. In
wooded landscapes, where North-American native species are more abundant, East-Asian pollen
was also overrepresented in larval provisions. In landscapes dominated by European cherry trees,
the bees collected from not only cherries, but European, North American, and Asian rosaceous
species evenly. Although we did not test plant species abundance, high dominance and low
diversity in each provision indicates foraging preference to a subset of host-plant species available.
These data indicate that East-Asian Rosaceae species are indeed preferred by *O. cornifrons*[14],
but they will use non-Asian rosaceous host-plants based on local floral availability[25,28,41]
(Figure 1,2). *Osmia cornifrons* did collect non-Asian, non-rosaceous pollen, as indicated by apple
and suburban sites (Figure 1). The use of non-preferred hosts may be the result of a dearth of
rosaceous pollen post-apple bloom in orchards; or that alternative pollen resources are needed in
their diet. Yet how bees adjust their foraging behavior in absence of preferred resources warrants
further investigation (see [22,42]).

The ability to use host-plant species within phylogenetic preferences supports that mesolectic
pollinators are ideal for crop pollination[17]. In apple orchards, the commercial apple (*Malus*
*pumila*) and its East-Asian pollinizer crabapple (*M. baccata*) were represented evenly in pollen
loads, indicating that *O. cornifrons* is delivering pollination services[12]. Cherries represent non-
native host-plant species. We did not observe the same degree of preference for *Prunus* pollen in
cherry orchards as we observed for *Malus* pollen in apple orchards (Figure 2), further supporting
the native preference hypothesis. The phylogenetic affinity of mesolectic bees for host plant
families that have cosmopolitan distributions has been exploited in other bee species, such as
*Megachile rotundata*, which is a managed pollinator for a variety of fabaceous crops[13].

Risks associated with the introduction of exotic bee species include pollination of invasive plant
species and competition with native bee species [14-18]. Although *O. cornifrons* collected locally
native pollen (e.g. *Acer*, *Amelanchier*, *Cercis*, *Rubus*); they also collected from invasive species
and common weeds of natural and agricultural habitats (e.g. *Barbarea*, *Elaeagnus*, *Rosa*,
*Taraxacum*)[14]. However, our results suggest that *O. cornifrons* may not be directly competing
for floral resources with other native *Osmia* spp.[43,44] because of *O. cornifrons*' preference for
East-Asian species. The Eastern North American native *O. lignaria lignaria* prefers early
blooming trees such as *Quercus* and *Salix*[45]. Western *O. californica* exhibit preference for
*Hydrophyllum*, and *O. l. propinqua* for *Hydrophyllum*, *Salix*, and the widespread
*Crataegus*[46,47]. These preferred pollen hosts from North American *Osmia* spp. are absent from
our dataset. Potential competitive foraging between *O. cornifrons* and *O. l. lignaria* on *Cercis* spp.

could occur[45], but *O. cornifrons* overall preference for Rosaceae may decrease this
probability[20,22,24,44,45]. However, specific and detailed studies would need to determine the
level and effect of competition between species[15,17,44] (including competition for nesting
sites).

Datasets like these can provide insights for appropriate management practices. For instance,
managing mesolectic pollinators that prefer pollen from crop plants allows growers to place nests
at long distances from field edges where pollination services from wild bees are generally
reduced[25,29-31,41]. However, bees could be limited to short blooming periods and low
nutritional diversity of the crops (such as orchard species), where they still need phenological and
phylogenetic diversity of pollen species over their active period (see Figure 1 for apple orchards)
[23,48]. By understanding bees' host-plant preferences, we can recommend complementary native
edge and hedgerow host-plant species to support bee populations (e.g. *Rubus* spp. and *C.*
*canadaensis* for *O. cornifrons* in orchards)[22,23,49,50].

Over 80 bee species have been intentionally or accidentally introduced outside of their native range
globally, yet how introduced bees adapt to floral communities outside of their native range is only
known for pollen generalist honey bees and bumble bees[15,17]. Our data supports that pollinators
do show affinity to plants from their native range with and will seek these pollen sources in new
habitats[14]. This study reveals that despite preferences for a subset of plant families from their
native range as pollen sources, mesolectic bees may successfully establish in new communities
when put in contact with novel related host-plants from other geographic origins. Mesolectic
foraging provides a balance between specificity to wild and targeted crop species and the ability
to utilize phylogenetically related pollen sources within their diet breadth constraint. This foraging
flexibility may be due to phylogenetically conserved floral reward quality[51,52] and bee
physiology[53-55], shared phenology[43], and bee behavior and floral morphology[3] allowing
bees plasticity in local diet choice. The globalization of cultivated crop, ornamental, and invasive
plant species may further facilitate the naturalization of introduced pollinators in novel
environments[56].

**Figure captions**

**Figure 1.** *Osmia cornifrons* larval provision host-plant genera by landscape colored by plant
family. Data in left panel are mean normalized count \pm SE. Boxes outlined in black are East-Asian
origin. Right panel indicates proportions of pollen collected from East-Asian and/or
Rosaceae/Fabaceae pollen vs not. Note that Rosaceae genera are the most represented across all
landscapes and East-Asian abundances vary by landscape.

**Figure 2.** Rosaceae plant species in *Osmia cornifrons* larval provisions by landscape. Data are
mean normalized count \pm SE. Box color represents target orchard crops of the particular landscape
(green), non-target crops (blue), or not crop species (burgundy); boxes outlined in black are East-

241 Asian origin.

**Funding**

This work was supported by the USDA-SCRI grant PEN04398, USDA-SCRI Coordinated
Agricultural Project grant MICL05063, USDA Hatch Appropriations under Project PEN04619,
and the State Horticultural Association of Pennsylvania.

**Permission to carry out fieldwork**

No permissions were required prior to conducting field work; all sites are privately owned and
work in coordination with Penn State Fruit Research and Extension Center.

**References**

- 1. Ollerton J, Winfree R, Tarrant S. How many flowering plants are pollinated by animals?
*Oikos*. 2011;120(3):321-6.
- 2. Cane JH, Sipes S. Characterizing floral specialization by bees: analytical methods and a
revised lexicon for oligolecty. *Plant-pollinator interactions: from specialization to generalization*.
2006:99-122.
- 3. Willmer P. *Pollination and Floral Ecology*: Princeton University Press; 2011. 778 p.
- 4. Anderson B, Johnson SD. The geographical mosaic of coevolution in a plant-pollinator
mutualism. *Evolution*. 2008;62(1):220-5.
- 5. López-Uribe MM, Cane JH, Minckley RL, Danforth BN. Crop domestication facilitated
rapid geographical expansion of a specialist pollinator, the squash bee *Peponapis pruinosa*.
*Proceedings of the Royal Society B-Biological Sciences*. 2016;283(1833):20160443.
- 6. Khoury CK, Achicanoy HA, Bjorkman AD, Navarro-Racines C, Guarino L, Flores-
Palacios X, et al. Origins of food crops connect countries worldwide. *Proceedings of the Royal*
*Society B-Biological Sciences*. 2016;283:20160792.
- 7. Garibaldi LA, Steffan-Dewenter I, Winfree R, Aizen MA, Bommarco R, Cunningham
SA, et al. Wild pollinators enhance fruit set of crops regardless of honey bee abundance. *Science*.
2013;339(6127):1608-11.
- 8. Junqueira CN, Augusto SC. Bigger and sweeter passion fruits: effect of pollinator
enhancement on fruit production and quality. *Apidologie*. 2017;48(2):131-40.
- 9. Bobiwash K, Uriel Y, Elle E. Pollen foraging differences among three managed
pollinators in the highbush blueberry (*Vaccinium corymbosum*) Agroecosystem. *Journal of*
*Economic Entomology*. 2018;111(1):26-32.
- 10. Batra SWT. *Anthophora pilipes villosula* SM (Hymenoptera, Anthophoridae), a
manageable Japanese bee that visits blueberries and apples during cool, rainy, spring weather.
*Proceedings of the Entomological Society of Washington*. 1994;96(1):98-119.
- 11. Batra SWT. Solitary bees for *Vaccinium* pollination. *Sixth International Symposium on*
*Vaccinium Culture*. 1997(446):71-6.
- 12. Bosch J, Kemp WP. Developing and establishing bee species as crop pollinators: the
example of *Osmia* spp. (Hymenoptera : Megachilidae) and fruit trees. *Bulletin of Entomological*
*Research*. 2002;92(1):3-16.
- 13. Pitts-Singer TL, Cane JH. The alfalfa leafcutting bee, *Megachile rotundata*: the world's
most intensively managed solitary bee. In: Berenbaum MR, Carde RT, Robinson GE, editors.
*Annual Review of Entomology, Vol 56. Annual Review of Entomology. 56. Palo Alto: Annual*
*Reviews*; 2011. p. 221-37.

14. Cane J. Exotic nonsocial bees (Hymenoptera: Apiformes) in North America: Ecological
implications. In: Strickler K, Cane J, editors. For Nonnative Crops, Whence Pollinators of the
Future? Lanham, MD: Entomological Society of America; 2003. p. 113-26.
- 15. Goulson D. Effects of introduced bees on native ecosystems. *Annual Review of Ecology,*
*Evolution, and Systematics.* 2003;34:1-26.
16. Hedtke SM, Blitzer EJ, Montgomery GA, Danforth BN. Introduction of non-native
pollinators can lead to trans-continental movement of bee-associated fungi. *PLoS ONE.*
2015;10(6):e0130560.
- 17. Russo L. Positive and negative impacts of non-native bee species around the world.
*Insects.* 2016;7(4):69-22.
18. Mallinger RE, Gaines-Day HR, Gratton C. Do managed bees have negative effects on
wild bees?: A systematic review of the literature. *PLoS ONE.* 2017;12(12):e0189268.
19. McKinney MI, Park Y-L. Nesting activity and behavior of *Osmia cornifrons*
(Hymenoptera: Megachilidae) elucidated using videography. *Psyche: A Journal of Entomology.*
2012;2012(6):1-7.
- 20. Biddinger D, Ngugi H, Frazier J, Frazier M, Leslie T, Donovall L. Development of the
mason bee, *Osmia cornifrons*, as an alternative pollinator to honey bees and as a targeted
delivery system for biological control agents in the management of fire blight. *Penn Fruit News.*
2010;90:35-44.
21. Park MG, Joshi NK, Rajotte EG, Biddinger DJ, Losey JE, Danforth BN. Apple grower
pollination practices and perceptions of alternative pollinators in New York and Pennsylvania.
*Renewable Agriculture and Food Systems.* 2018;129:1-14.
- 22. Haider M, Dorn S, Sedivy C, Muller A. Phylogeny and floral hosts of a predominantly
pollen generalist group of mason bees (Megachilidae: Osmini). *Biological Journal of the*
*Linnean Society.* 2014;111(1):78-91.
23. Nagamitsu T, Suzuki MF, Mine S, Taki H, Shuri K, Kikuchi S, et al. Effects of forest loss
and fragmentation on pollen diets and provision mass of the mason bee, *Osmia cornifrons*, in
central Japan. *Ecological Entomology.* 2018;43(2):245-54.
- 24. Russo L, Danforth B. Pollen preferences among the bee species visiting apple (*Malus*
*pumila*) in New York. *Apidologie.* 2017;48(6):806-20.
25. Biddinger DJ, Joshi NK, Rajotte EG, Halbrecht NO, Pulig C, Naithani KJ, et al. An
immunomarking method to determine the foraging patterns of *Osmia cornifrons* and resulting
fruit set in a cherry orchard. *Apidologie.* 2013;44(6):738-49.
26. Fridley JD. Of Asian forests and European fields: Eastern U.S. plant invasions in a global
floristic context. *PloS one.* 2008;3(11):e3630.

27. Ricklefs RE, Guo QF, Qian H. Growth form and distribution of introduced plants in their
native and non-native ranges in Eastern Asia and North America. *Diversity and Distributions*.
2008;14(2):381-6.
28. Kammerer MA, Biddinger DJ, Rajotte EG, Mortensen DA. Local plant diversity across
multiple habitats supports a diverse wild bee community in Pennsylvania apple orchards.
*Environmental Entomology*. 2016;45(1):32-8.
29. Gathmann A, Tschardt T. Foraging ranges of solitary bees. *Journal of Animal Ecology*.
2002;71(5):757-64.
30. Guedot C, Bosch J, Kemp WP. Relationship between body size and homing ability in the
genus *Osmia* (Hymenoptera; Megachilidae). *Ecological Entomology*. 2009;34(1):158-61.
31. Zurbuchen A, Landert L, Klaiber J, Muller A, Hein S, Dorn S. Maximum foraging ranges
in solitary bees: only few individuals have the capability to cover long foraging distances.
*Biological Conservation*. 2010;143(3):669-76.
32. McKinney MI, Park Y-L. Distribution of *Chaetodactylus krombeini* (Acari:
*Chaetodactylidae*) within *Osmia cornifrons* (Hymenoptera: Megachilidae) nests: implications for
population management. *Experimental and Applied Acarology*. 2012;60(2):153-61.
33. Keller A, Danner N, Grimmer G, Ankenbrand M, von der Ohe K, et al. Evaluating
multiplexed next-generation sequencing as a method in palynology for mixed pollen samples.
*Plant Biology*. 2015;17(2):558-66.
34. Sickel W, Ankenbrand MJ, Grimmer G, Holzschuh A, Härtel S, Lanzen J, et al. Increased
efficiency in identifying mixed pollen samples by meta-barcoding with a dual-indexing
approach. *BMC ecology*. 2015;15(1):20.
35. Keller A, Hohlfeld S, Kolter A, Schultz J, Gemeinholzer B, Ankenbrand MJ.
BDdatabaser: On-the-fly reference database creation for (meta-)barcoding. *EcoEvoRxiv*. 2019.
doi: 10.32942/osf.io/cmfu2
36. Rognes T, Flouri T, Nichols B, Quince C, Mahe F. VSEARCH: a versatile open source
tool for metagenomics. *Peerj*. 2016;4:22.
37. McMurdie PJ, Holmes S. phyloseq: An R Package for Reproducible interactive analysis
and graphics of microbiome census data. *PLoS One*. 2013;8(4):11.
38. Deagle BE, Thomas AC, McInnes JC, Clarke LJ, Vesterinen EJ, Clare EL, et al.
Counting with DNA in metabarcoding studies: How should we convert sequence reads to dietary
data? *Molecular Ecology*. 2019;28(2):391-406.
39. Lamb PD, Hunter E, Pinnegar JK, Creer S, Davies RG, Taylor MI. How quantitative is
metabarcoding: A meta-analytical approach. *Molecular ecology*. 2019;28(2):420-30.

40. Pornon A, Baksay S, Escaravage N, Burrus M, Andalo C. Pollinator specialization
increases with a decrease in a mass-flowering plant in networks inferred from DNA
metabarcoding. *Ecology and evolution*. 2019;279(12):4845-13.
- 41. Joshi NK, Otieno M, Rajotte EG, Fleischer SJ, Biddinger DJ. Proximity to woodland and
landscape structure drives pollinator visitation in apple orchard ecosystem. *Frontiers in Ecology
and Evolution*. 2016;4:142.
- 42. Kratschmer S, Petrovic B, Curto M, Meimberg H, Pachinger B. Pollen availability for the
Horned mason bee (*Osmia cornuta*) in regions of different land use and landscape structures.
*Ecological Entomology*. 2019;10.1111/een.12823
- 43. Raw A. Pollen Preferences of Three *Osmia* Species (Hymenoptera). *Oikos*.
1974;25(1):54.
- 20 366 44. Centrella ML. Regional and Local Drivers of Mason Bee (Genus *Osmia*) Decline Across
21 367 the Eastern Seaboard. Ann Arbor: Cornell University; 2019.
- 45. Kraemer ME, Favi FD. Flower phenology and pollen choice of *Osmia lignaria*
(Hymenoptera: Megachilidae) in Central Virginia. *Environmental Entomology*.
2005;34(6):1593-605.
- 46. Williams NM, Tepedino VJ. Consistent mixing of near and distant resources in foraging
bouts by the solitary mason bee *Osmia lignaria*. *Behavioral Ecology*. 2003;14(1):141-9.
- 47. Levin MD. Biological notes on *Osmia lignaria* and *Osmia californica* (Hymenoptera:
Apoidea, Megachilidae). *Journal of the Kansas Entomological Society*. 1966;39(3):524-35.
- 48. Vaudo AD, Tooker JF, Grozinger CM, Patch HM. Bee nutrition and floral resource
restoration. *Current Opinion in Insect Science*. 2015;10:133-41.
- 49. Park M, Danforth B, Losey J, Biddinger D, Vaughan M, Dollar J, et al. Wild pollinators
of eastern apple orchards and how to conserve them: College of Agriculture and Life Sciences,
Cornell University; 2012.
- 50. Vaughn M, Lee-Mäder E, Cruz JK, Goldenetz-Dollar J, Gill K, Borders B. Hedgerow
planting (422) for pollinators: Pennsylvania installation guide and job sheet. Portland, OR: The
Xerces Society; 2015.
- 51. Vaudo AD et al. Pollen protein:lipid macronutrient ratios may guide broad patterns of bee
species floral preferences. *Insects*. 2020 (in press).
- 52. Ruedenauer FA, Spaethe J, van der Kooi CJ, Leonhardt SD. Pollinator or pedigree: which
factors determine the evolution of pollen nutrients? *Oecologia*. 2019:1-11.
- 53. Vaudo AD, Stabler D, Patch HM, Tooker JF, Grozinger CM, Wright GA. Bumble bees
regulate their intake of essential protein and lipid pollen macronutrients. *The Journal of
Experimental Biology*. 2016;219(Pt 24):3962-70.

54. Vaudo AD, Patch HM, Mortensen DA, Tooker JF, Grozinger CM. Macronutrient ratios
in pollen shape bumble bee (*Bombus impatiens*) foraging strategies and floral preferences.
Proceedings of the National Academy of Sciences. 2016;113(28):E4035-42.
55. Giri S, Rule DC, Dillon ME. Fatty acid composition in native bees_ Associations with
thermal and feeding ecology. Comparative Biochemistry and Physiology, Part A. 2018;218:70-9.
56. Brown J, Cunningham SA. Global-scale drivers of crop visitor diversity and the historical
development of agriculture. Proceedings of the Royal Society Biology B: Biological Sciences.
2019;286:20192096.

Figure 1. *Osmia cornifrons* larval provision host-plant genera by landscape colored by plant family. Data in left panel are mean normalized count \pm SE. Boxes outlined in black are East-Asian origin. Right panel indicates proportions of pollen collected from East-Asian and/or Rosaceae/Fabaceae pollen vs not. Note that Rosaceae genera are the most represented across all landscapes and East-Asian abundances vary by landscape.

564x436mm (300 x 300 DPI)

Figure 2. Rosaceae plant species in *Osmia cornifrons* larval provisions by landscape. Data are mean normalized count \pm SE. Box color represents target orchard crops of the particular landscape (green), non-target crops (blue), or not crop species (burgundy); boxes outlined in black are East-Asian origin.

431x356mm (300 x 300 DPI)

Appendix B

Dear Dr Vaudo,

The editors assigned to your paper ("Phylogenetic pollen preferences facilitate optimal pollination services and may ease the establishment of introduced bees in new habitats") have now received comments from reviewers. We would like you to revise your paper in accordance with the referee and Associate Editor suggestions which can be found below (not including confidential reports to the Editor). Please note this decision does not guarantee eventual acceptance.

Please submit a copy of your revised paper before 28-Jun-2020. Please note that the revision deadline will expire at 00.00am on this date. If we do not hear from you within this time then it will be assumed that the paper has been withdrawn. In exceptional circumstances, extensions may be possible if agreed with the Editorial Office in advance. We do not allow multiple rounds of revision so we urge you to make every effort to fully address all of the comments at this stage. If deemed necessary by the Editors, your manuscript will be sent back to one or more of the original reviewers for assessment. If the original reviewers are not available, we may invite new reviewers.

- Data accessibility

<http://datadryad.org/submit?journalID=RSOS&manu=RSOS-200225>

- Competing interests

- Authors' contributions

- Acknowledgements

- Funding statement

on behalf of Dr Sean Rands (Associate Editor) and Pete Smith (Subject Editor)
openscience@royalsociety.org

Associate Editor's comments (Dr Sean Rands):

Associate Editor: 1

Comments to the Author:

Thank you for submitting your manuscript to Royal Society Open Science. Two referees have now provided their comments on your manuscript.

Please ensure that you address their comments fully while revising your manuscript, and

provide a point-by-point response upon resubmission.

Comments to Author:

Reviewers' Comments to Author:

Reviewer: 1

Comments to the Author(s)

General comments:

In this study, Vaudo et al. examine the foraging preferences of an introduced pollinator, the solitary bee *Osmia cornifrons*, in North American habitats. The hypotheses they propose are that hosts that have co-evolved with *O. cornifrons* will be more attractive than non-native hosts, and that *O. cornifrons* will preferentially forage on species related to their native hosts compared to unrelated hosts. In their study, they placed *O. cornifrons* nests at four location types – apple orchards (one of their native hosts), cherry orchards (same family as native host), a suburban site and a woodland forest. They then used ITS2 metabarcoding on the pollen nest provisions to work out the relative quantities of each pollen type, thus examining the foraging choice of *O. cornifrons* when presented with a range of potential hosts.

The authors have presented a novel and interesting application of pollen metabarcoding, by testing whether there is a native host preference and/or phylogenetic influence on the foraging preferences of *O. cornifrons* in introduced habitats, and how this may influence their ability to adapt outside of their native range. This is a great question that has economically important implications to crop pollination management. For example, can we predict or manage the foraging preferences of introduced pollinators?

The authors found that, in most landscapes, *O. cornifrons* collect significantly more pollen from rosaceous species originating from their native East Asian range. Their study also showed that, when placed at the edge of a rosaceous crop field, *O. cornifrons* is pollinating those crops as anticipated (more so in apple orchards than cherry). As the authors point out, these results provide valuable insights to agricultural managers. The results from the suburban and woodland sites were also very interesting, as mass-flowering crops, I assume, were not within foraging range. In these two sites, they found that *O. cornifrons* will preferentially forage on non-East Asian Fabaceae, but on woodland sites they preferentially forage on both native and non-native Rosaceae. Both of these families have been identified as preferred hosts in their native range, providing support to the hypothesis that the relatedness to native hosts influences the bees' foraging preference.

From the information provided, I wondered if we are seeing an effect of resource availability – bees are merely foraging on what is closest to their nests. It would be good to know more about the composition of vegetation within foraging range, and whether alternative hosts were fully flowering at the time. For example, how close are apple trees to the cherry orchard (and conversely, cherry trees to apple orchards)? *Malus* are clearly within the foraging range of the bee nesting next to cherry orchards, as *Malus* pollen was found in those provisions. Other points of discussion could include: are foraging patterns likely to be different at different times of the year when alternative hosts are flowering? Would placing nests near a crop from a different family (non-rosaceous) yield a dominance of pollen from the unrelated host, or would they still seek out rosaceous hosts?

- We appreciate the comments of the reviewer here. Clearly there are site level differences in plant species found in larval provisions. However, the consistency of our results with published literature of *O. cornifrons* for family level preferences, indicate that the bees' foraging behavior is not simply driven by the dominance of a

plant in the landscape, because there indeed were more flowering plants in the landscape at the time of foraging. Given this, our goal in this study was not to test the effect of resource abundance on the bees' preference but whether there was a signal of preferences based on realistic field situations (especially because quantifying pollen quantity and abundance among all host plants within a liberal estimated foraging range of the bee is a separate question altogether). A controlled experiment to test this would be necessary. Nonetheless we added some more commentary in the discussion about this

- lines 203-222: "Our results indicate that *O. cornifrons* family level host-plant preferences drive their foraging behavior, corroborating that this is a mesolectic species that specializes on a few plant families: Rosaceae and Fabaceae [2]. Given *O. cornifrons* limited foraging range, the variation between sites that we detected is probably driven by local availability of host-plant species. For instance, where *Cercis* was planted in high abundance in suburban landscapes, opposed to orchards, we found this type of pollen in high abundance. We did not directly measure flowering plant species abundance at each site during *O. cornifrons* foraging period, but replication across a variety of sites reveals similar foraging trends. Many plant species from multiple families were blooming at the same time (Table S1, Figure S4, and see [28]), including mass blooming trees such as *Acer* (Sapindaceae) or *Salix* (Salicaceae), which were found in low abundance or not at all in our samples. Additionally, low diversity of larval provisions (composed of multiple foraging trips) indicates *O. cornifrons* forages to a subset of host-plant species available (Figure S2,S4). Our data indicate that East-Asian Rosaceae species are indeed preferred by *O. cornifrons* [14], but they will use non-Asian Rosaceae host-plants based on local floral availability [25,28,43] (Figure 1,2). The use of non-Asian host plants may be the result of their sampling the landscape, a byproduct of nectar foraging as opposed to active pollen foraging, a dearth of Rosaceae and Fabaceae pollen post-crop bloom in orchards, or that alternative pollen resources are needed in their diet. How bees adjust their foraging behavior in absence of preferred resources, or the effects of local floral abundance on species preferences warrants further investigation in more controlled settings (see [22,44])."

Another comment is that there are some questions regarding the quantitative potential of metabarcoding, as discussed in the references cited (Deagle et al., Pornon et al., etc), and, more recently, by Bell et al., 2019 and Richardson et al., 2019. A mention of these issues and their possible impact on the interpretation of the results should be included in the discussion.

- We recognize that the methods and quantitative potential of metabarcoding is still being assessed with multiple techniques. On a sample-by-sample basis, there may be some inaccuracies in absolute abundance of pollen in each sample. However, we analyzed relative read abundance data in a population-wide sense, as Deagle et al., 2019 have shown to be suitable. Therefore, because we analyze our results at the site and landscape level, and thus the population level, we believe the interpretation of our results are valid. Because we wanted our discussion to focus on the ecological application of the method, and debating the quantitative potential of this method is outside the scope of our study, we elaborated more on our justification to use RRAs in the Materials and Methods
- lines 109-116: "Because relative pollen abundances (RRA) by light microscopy and ITS2 metabarcoding were previously strongly correlated with the primers applied here [33], we used RRA as a proxy for pollen abundance in larval provisions [38-40]. Because we were analyzing bee population foraging trends in different habitats, we further analyzed RRAs by site or landscape using the mean of RRAs from 12 samples per site. We thus use

RRAs as a population wide estimate of pollen use and not as sample-wise abundance estimations [38,41,42] (see table S1 for species list and metadata and for raw data see <https://doi.org/10.5061/dryad.ffbg79cqn>).”

Overall, the authors have explored an important ecological question and have provided good evidence of foraging preferences of *O. cornifrons* under some typical agricultural scenarios using novel molecular methods. I found the manuscript clearly written and enjoyable to read, and it should appeal to a broad audience. I think the manuscript would further benefit from more details about the main concepts being explored (i.e. what we expect under mesolectic vs generalist vs oligolectic foraging preferences) and of the experimental design, as well as a broader discussion about other factors that may be influencing the results observed.

Minor comments:

Line 26. A more detailed explanation of the terms oligolecty and mesolecty and their use in this study would be useful for non-expert readers.

- In our edits, we removed the terms here, but included a definition for mesolecty in lines 42-45: “It has been demonstrated that *O. cornifrons* populations in Japan, Russia, and North America exhibit pollen fidelity to Rosaceae and Fabaceae species [22-24], indicating it may be a mesolectic forager (collecting pollen from many species within 1-3 main plant families [2]).”

Line 62. How many nests in total were placed at each site?

- We included that we placed five boxes at each site

Line 72. How many nests in total were sampled? How long do the nest provisions take to collect? – this would help us understand over what time period the bees foraged for the pollen provisions.

Line 73. Why are nests collected from nests where eggs failed to hatch (is it because they are intact)? Perhaps clarify this.

- To clarify these two comments, we added the following information
- lines 72-78: “All females were allowed to forage for their entire active period (approximately four weeks) where one to two cells are provisioned per day in ideal temperature and precipitation conditions [21]. At the end of the growing season, we collected 12 complete larval provisions from cells of different individual nests per site (N=96) where eggs failed to hatch (and were not parasitized by mites [32]). These pollen provisions were selected because they were unused and complete and therefore constitute all pollen provisioned for that cell.”

Line 77. “spiked with custom index, Read1, and Read2 primers”. Could you clarify what the purpose of this spike-in was? Or should it read 5% PhiX spiked in to the pooled library?

- We apologize for the confusion here. The MiSeq reagent cartridge of course already contains the standard Illumina Read1, Read2 and Index Read Primers. However, we used a custom protocol with custom primer sequences, where the Illumina primers would not work. The custom sequencing primers were spiked into the MiSeq reagent cartridge, into wells 12, 13 and 14. There are optional wells for custom protocols (18, 19, 20) but these wouldn't help in our case, because we also have PhiX, which we need for sequence quality - and for these sequences, the standard Illumina primers apply. We adjusted this in lines 85-88:

- "The final pooled library was spiked with 5% PhiX control to increase sequencing quality. The library was sequenced on an Illumina MiSeq v2 2x250bp spiked with custom index, Read1, and Read2 sequencing primers to bind to the unique ITS2 primers [34]."

Line 84. Were there any negative controls included that helped inform subsequent filtering?

- We used negative controls in PCR to visualize that there was no contamination, but they were not included in the pooled library sent for sequencing. We included that we used negative controls
- lines 83-84: "Negative controls were used in PCR to verify there was no plant DNA contamination."

Line 88. Reference for JMP Pro please.

- We added references for JMP and PRIMER

Line 116. Some statistics on total sequencing reads/reads per sample would be useful.

Where there many unidentified/unassigned reads?

- We added these stats in the manuscript
- lines 95-96: "Across all 96 samples, we obtained a total of 6,800,673 reads passing our quality filters, classified 4,922,865 reads, and averaged $51,279.84 \pm 4872.85$ SE classified reads per sample."

Line 117. Does dominance here refer to Simpson dominance index, rather than dominance in the physical sense (e.g. Line 51)? Also please clarify how the results support mesolectic foraging – what would be expected under oligolectic foraging or another scenario?

- Yes this refers to Simpson's index. We clarified this and how it indicates mesolectic foraging in lines 145-152. Additionally we distinguished between Simpson's dominance and physical abundance of plant species throughout the manuscript:
- Lines 145-152: "*Osmia cornifrons* pollen provisions comprised 5-8 species, 1-6 genera, and 4 families on average. Simpson's dominance averaged 0.45 at the genus level (indicating low diversity), and that this species is indeed a mesolectic forager [2] (i.e. as their collections were dominated plant species within few plant families [2,28]). Diversity and Simpson's dominance did not differ between landscapes, except species richness was highest in cherry orchards (Figure S2; Species Richness)."

Line 124. By species analyzed, do you mean detected in metabarcoding?

- That is correct; we corrected this to say "species we detected with metabarcoding"

Line 131. In the urban scenario, how abundant are Fabaceae species (*Cercis*)?

- It was much more abundant than in agricultural landscapes or in the woods, we addressed this in an added section in the discussion regarding landscape context
- lines 160-163: "This was consistent across all sites and landscapes except the suburban site where *Cercis* (Fabaceae) pollen was frequently collected, likely due to the high density of ornamental planting of *C. canadensis* in the urban areas compared to agricultural and wild landscapes. (Figure 1,S4)."
- Lines 210-212: "For instance, where *Cercis* was planted in high abundance in suburban landscapes, opposed to orchards, we found this type of pollen in high abundance."

Line 141. Does "redundancy" here refer to reporting of results? (i.e. to reduce the number of times significance is mentioned?)

- This is correct, we changed the phrase

- lines 141-143: "To reduce repeating significance values, all reported results are significant at $P < 0.05$; χ^2 values are reported for contingency analysis unless GLM is indicated."

Line 147. The authors mention that alternative host plants are abundant; however, it is not clear how this was determined as we don't have information regarding the composition of vegetation at all sites. Were they flowering in similar numbers to the crop plants?

- We did not sample the pollen availability and abundance of all flowering plant species at all sites and cannot answer this question quantitatively. However, the majority of bees do not forage haphazardly nor simply forage at the same frequency of pollen availability, and *O. cornifrons* shows family level preferences across studies. We know that the bees did not forage on some species that were clearly available (Kammerer et al 2016 published vegetation among some of the same apple orchards). We did however add some discussion about justification of our conclusions in the discussion. Please see response to first major comment above.

Line 151. "high representation rosaceous pollen (found in 96% of samples)". Suburban samples had a low representation of rosaceous pollen, but they were few in number, so this comparison is somewhat biased as half of the sites were next to rosaceous crops.

- We clarified this line to mean that we found rosaceous pollen in 96% of the samples, not that there was an average of 96% rosaceous pollen per sample.
- lines 179-182: "Overall, *O. cornifrons* foraging is mostly driven by phylogenetic pollen preferences [2,14] showing high representation rosaceous pollen (detecting Rosaceae species in 96% of all samples) of East-Asian, European and North American origin (Figure S4,S5)."

Line 199. Were *Crataegus* and *Salix* flowering at the time this study was conducted?

- They do in fact bloom during *O. cornifrons* foraging period which we added.
- lines 252-253: "These preferred pollen hosts from North American *Osmia* spp. are absent from our dataset even though they bloom during *O. cornifrons* foraging period [28]."

Figure 1. Could you perhaps split Rosaceae and Fabaceae contributions in the right panel to match the left panel?

- We do appreciate the suggestion here, but splitting the right panel further seems to make the figure more confusing, as this was designed to show the interaction between our 2 hypotheses from the introduction. The left panel should convey the genera breakdown of the different families. We decided to leave it as is for now.

Figure S2. Consider coloring or separating the different site types to make the figure easier to interpret.

- We changed the colors such that each site was associated with habitat type.

Figure S4. What do mean normalized counts refer to? The number of reads?

- This is correct, we changed all figure captions and descriptions in text to "relative read abundances (RRA)" as we corrected in the methods (see lines 90-116)

Data files. Could the taxonomy database file (its2.penn-malus2018-5.fasta) be included?

- We included the database file in the dryad data

Reviewer: 2

Comments to the Author(s)

RSOS-200225. Vaudo et al. Phylogenetic pollen preferences facilitate optimal pollination services and may ease the establishment of introduced bees in new habitats

This work examines whether the solitary bee *Osmia cornifrons* shows a pollen-collecting preference for co-evolved plants from its native range and/or a preference for a subset of phylogenetically related plants. The preferences were tested in apple, cherry, woodland and suburban environments.

The manuscript is well laid out, generally clearly written, and brief. This is an interesting topic and I enjoyed reading the manuscript. I've made some comments directly in the pdf.

- We exported the comments from the pdf and addressed them one-by-one below

While I note the aims of the work (per the abstract), overall, I found the hypothesis and conclusions somewhat unclear and slightly frustrating. The 'and/or' nature of the aims perhaps contributes to this. For phylogenetic preference, the authors note that *O. cornifrons* is mesolectic for Rosaceae and Fabaceae (L37/38). Was this in doubt? Was there – for example - any question that *O. cornifrons* would collect cherry pollen, in preference to non Rosaceae/Fabaceae alternatives? If there was room for discovery in this area, it would be useful if this was more specifically outlined.

- We apologize the reviewer was confused by our hypotheses and we made some changes throughout the introduction to clarify these (see lines 23-33, and 50-62). The "doubt" the reviewer mentioned is in the species level preferences of the bee. Using metabarcoding, it would be the first time that we can analyze this. Previous studies only addressed preferences at the family (or genus) level and so therefore, we could not test the species source of the pollen which helps us address pollination, nectar vs pollen foraging, or in this case, testing the native preference hypotheses.
- We tried to clarify this in lines 42-48: "It has been demonstrated that *O. cornifrons* populations in Japan, Russia, and North America exhibit pollen fidelity to Rosaceae and Fabaceae species [22-24], indicating it may be a mesolectic forager (collecting pollen from many species within 1-3 main plant families [2]). However, these previous studies analyzed pollen via microscopy which only indicated pollen family or genus. Therefore, data are absent regarding the species-level identity of the pollen preferences of *O. cornifrons*, and the degree of pollen collected from native and introduced host-plants species in different landscapes [22-25]."

With less unknowns (I think) with respect to phylogenetic preference, the focus was seemingly on the tricky task of identifying native range preferences in the field. How to separate these preferences from attendance at the dominant available pollen source? It is difficult for me to see how the native range hypothesis would be properly tested unless the bees were placed in proximity to closely related plant species (such as *Malus*?) of different geographical origin. Some quantification of the available pollen resources would be useful and it's unclear to me if the authors tried to do this? It certainly isn't prominently described.

- Here our goal was to investigate the use of pollen by estimating abundance in larval provisions in realistic settings, rather than giving the bees foraging choices in a controlled experiment. Given that, we changed phrases throughout the manuscript that might imply we were testing preferences. We addressed this further in the discussion. See our next response

The authors strongly suspect that *O. cornifrons* will gather pollen from the dominant locally available Rosaceae and Fabaceae; "Therefore, we predict that *O. cornifrons* would exhibit foraging preference for East-Asian Rosaceae and Fabaceae in landscapes dominated by these plants (apple orchards and suburban areas). In contrast, we predict that *O. cornifrons* would forage more on European or North American rosaceous and fabaceous species in landscapes where these plants are more abundant (cherry orchards and wooded forests)" (L51-55). But the authors don't really explain how they will get around this problem. Finally, in the last final sentence of the Introduction, the authors seemingly hypothesise against the native range preference.

- Our attempt to get around this issue is to observe the bees' larval provisions across multiple cells per site and multiple sites. With that much variety, we were hoping to reveal some trends which we did in the bees mesolectic foraging. The rationale behind using non-apple landscapes was to look at the pollen use in landscapes not dominated by east-asian Rosaceae, such as cherry, woods, and suburban landscapes. We understand that a controlled setting would allow us to truly test individual foraging preferences, yet we observed differences in the use of pollen realistic situations for this bee. We recognize this limitation and try to address it in the discussion lines 203-222:
- "Our results indicate that *O. cornifrons* family level host-plant preferences drive their foraging behavior, corroborating that this is a mesolectic species that specializes on a few plant families: Rosaceae and Fabaceae [2]. Given *O. cornifrons* limited foraging range, the variation between sites that we detected is probably driven by local availability of host-plant species. For instance, where *Cercis* was planted in high abundance in suburban landscapes, opposed to orchards, we found this type of pollen in high abundance. We did not directly measure flowering plant species abundance at each site during *O. cornifrons* foraging period, but replication across a variety of sites reveals similar foraging trends. Many plant species from multiple families were blooming at the same time (Table S1, Figure S4, and see [28]), including mass blooming trees such as *Acer* (Sapindaceae) or *Salix* (Salicaceae), which were found in low abundance or not at all in our samples. Additionally, low diversity of larval provisions (composed of multiple foraging trips) indicates *O. cornifrons* forages to a subset of host-plant species available (Figure S2,S4). Our data indicate that East-Asian Rosaceae species are indeed preferred by *O. cornifrons* [14], but they will use non-Asian Rosaceae host-plants based on local floral availability [25,28,43] (Figure 1,2). The use of non-Asian host plants may be the result of their sampling the landscape, a byproduct of nectar foraging as opposed to active pollen foraging, a dearth of Rosaceae and Fabaceae pollen post-crop bloom in orchards, or that alternative pollen resources are needed in their diet. How bees adjust their foraging behavior in absence of preferred resources, or the effects of local floral abundance on species preferences warrants further investigation in more controlled settings (see [22,44])."

While these points above are largely about the set-up of the manuscript, they are then, naturally, reflected in the Results and Discussion. There is a lack of firmness and structure about what was tested and what was concluded. Rather, the observations of pollen collected - in very different landscapes - are correlative/indicative (see the title itself).

In a study of native range preferences, taxonomic IDs to species level (at least best matches to species) must be important. Averaged details of the pollen IDs appear in the first line of the results, including a mention of species detected. However, the results for the species-level identifications are very scant. It may be that this information is given in the Dryad archive (which appears to be extensive), but this is not mentioned/cited in the text, and I cannot access the Dryad material. I think the authors really should present a table or

diagram that summarises the species-level findings in the text (or supp mat).

Why is it necessary to present Fig 1 at genus level? Is this because the authors are not confident of the ID to species level? Is there any possibility that a genus-level category might contain more than one species? It is noted at L182, pollen from both apple and crabapple were detected in the apple orchards. What and where were the *Malus* and the *Rubus* sp in the cherry orchard? I found the methods used for sequence processing and taxonomic identification to be slightly opaque (comments on L79-93). As one small point, the authors state that species identifications were verified, but I'd prefer to know how these were verified. A presentation of the species-level results might help illuminate the methods used.

- We appreciate the reviewer's concern here. We did indeed present the species level ID and raw data in our dryad submission. But we do understand that those data aren't readily accessible from the ms. We decided to add a plant species list in the supplementary material with appropriate categories used in the analysis. We also added a supplementary figure (Figure S4) of species level relative read abundances. Furthermore, we added additional explanation as to why we used genus level analysis in the manuscript in lines 133-136. The idea is that although multiple species (of which we are confident in identification) may be included in each genus, analyzing at the species level could reduce the importance of that genus in pollen provisions, especially regarding mesolectic bees that forage from many species within few plant families. Therefore, they were summed up to the genus (or family) level but split between east-asian or not for the specific analysis. For instance, if bees collected from 3 plant species within a genus and each collected at 20%, the average for that genus would be 20% in provisions when analyzed at the species level, as opposed the reality that the genus is preferred with a total of 60% of the pollen provision.
- We added in lines 133-136: "These tests were conducted by summing data at the genus level to avoid negative bias, where the average genus representation with high counts is reduced by rare species (with low counts) within the genus."
- The *Malus* and *Rubus* species in cherry orchards requested by the reviewer are presented in Figure 2. Finally, we have specific sequence processing methods presented in the dryad depository; as this is not a methods paper, we did not feel it appropriate to add too much in terms of processing in the manuscript. We did make it clear that those steps are provided in the dryad data however (see specific response below).

Individual comments from pdf

Line 4: It's a matter of taste, but I found the wording of this sentence to be unnecessarily complex. It's a contrast with the rest of the writing.

- We attempted to simply this in lines 3-6:
- "An ideal pollinator comprises particular traits related to its host-plant species: timing its foraging period with blooming period, exhibiting floral constancy to guarantee pollen transfer between conspecific flowers, and displaying fidelity to the host-plant species over generations [2,3]."

Line 7: It would be useful if some example(s) was given here. What sort of traits?

- We added in lines 6-8:
- "Relationships between pollinators and their host-plant species in their native geographic ranges have evolved over thousands to millions of years, leading to coevolved (morphological, behavioral, chemical, and physiological) traits [3,4]."

Line 21: intentionally?

- corrected

Line 38: It might be useful to give some indication of the study locations used in these prior studies.

- We added the locations in lines 42-45:
- "It has been demonstrated that *O. cornifrons* populations in Japan, Russia, and North America exhibit pollen fidelity to Rosaceae and Fabaceae species [22-24], indicating it may be a mesolectic forager (collecting pollen from many species within 1-3 main plant families [2])."

Line 44: I would suggest breaking up this sentence. It is really quite hard to digest at once.

- Corrected

Line 50: This section through to the end is key, but the framing of the hypothesis is pretty vague.

- We reworded some of the paragraph to hopefully make this more clear in lines 50-62: "The establishment of *O. cornifrons*, and abundance of crop and wild plant species from East-Asia in North America [26, 27], makes this an ideal system to determine how floral preferences of recently introduced pollinators may shift in new environments. Using pollen metabarcoding, we analyzed pollen species identity and relative abundance in *O. cornifrons* larval pollen provisions. To test the native hypothesis, we compared *O. cornifrons* pollen provisions between heterogeneous landscapes of exotic and native plants [28]: orchards with a high abundance of East-Asian crops (apples and pears), orchards with a high abundance of European crops (cherries), and landscapes with abundance of host-plants from different geographic origins (suburban and wooded forest). We predicted that if *O. cornifrons* prefers host plants from their native geographic range, we would find higher abundance of pollen from East-Asian Rosaceae and Fabaceae plants across all samples. We discuss our results in the context of how foraging plasticity to different species and abundant resources within host-plant family preferences can facilitate pollinator establishment in novel habitats."

Line 55: This statement seem to argue against a native host range preference. Is that the intention?

- We modified the presentation of our predictions, see the paragraph referenced in the preceding comment.

Line 82: Over the entire fragment? >97% and the top match?

How close were other sequences in the database?

- We clarified the method here in lines 92-94:
- "We classified reads by selecting the match with highest identity to a reference species over the entire amplified region with a global alignment using VSEARCH [36] and a threshold of at least >97% sequence identity."

Line 83: Is this really the best word to use?

- We've changed some words in the methods to refer to "relative read abundance"

Line 84: Less than 1% per sample or across the entire dataset (or per sample)?

- Per sample, which we added

Line 84: This isn't very clear. Were ASVs the basis for the original identification out of vsearch (if they were, why mention them in the 84-86 sentence, rather than 81-84)? Were the ASVs verified by alignment to Sanger sequences from Kammerer? How was the ID from this alignment judged to be different from/better than the first classification?

Did verification only occur for apple orchard samples, or did it extend to samples from the other environments?

- The ASVs were used to double check our original dataset. The data were consistent so we stuck with the original data. We visually inspected our plant list against that of Kammerer who sampled and botanically IDed plant species in apple orchards as a further verification that we were obtaining accurate species id. We tried to clarify this in lines 99-104:
- "We verified the consistency of species identification and abundance by comparing our results to amplicon sequence variants and therefore kept our original dataset for analysis. For classification and filtering scripts, refer to our dryad data entry (<https://doi.org/10.5061/dryad.ffbg79cqn>). We also verified consistency in species identification by comparing against the species characterized in the same apple orchards in Kammerer et al. 2016 [28]."

Line 88: more details?

- We added reference to species list and raw data at the end of the paragraph: "(see table S1 for species list and metadata and for raw data see <https://doi.org/10.5061/dryad.ffbg79cqn>)"

Line 89: So a second filter to reduce rare taxa? Site = sampling location? Would a genera be removed from the whole dataset (even if it was common from some sites?). Why are 'genera' mentioned here? Was that as far as tax ID went?

- Yes after original filtering there were certain taxa that were extremely rare, and not common among any site or sample. These were genera that totaled <0.3% per site across all sites. We specified this in the methods lines 109-110:
- "Final filtering removed plant genera that were extremely rare in the dataset, those that contained less than 0.3% of the total counts per site across all sites."

Line 124: Is this 73 plant species that were detected via the metabarcoding? This section talks about species identified but the Fig 1 only shows to genera level.

- Yes, detected via metabarcoding, we specified this in the line. We also added data for species in a supplementary material table, see above.

Line 124: I don't really see 'across all samples' as especially relevant. they are completely different landscapes. It could simply be the most abundant available pollen source. If the authors removed the "Yet", it would be a matter-of-fact result.

- Good point, we removed "yet"

Line 128: Was this predicted?

- Our prediction was landscape difference, but we removed the phrase for clarity.

Line 141: sounds quite M&M

- We moved this sentence to the conclusion of the methods.

Line 147: So they prefer non East-Asian Rosaceae? Or they are capable of utilising? How does this fit with L124-126?

- We rephrased this sentence in lines 176-178:

- "However, across sites, they broadly utilize pollen from Rosaceae species from outside their native range indicating their capability to utilize host-plants species they did not coevolve with (Figure 1,2)."

Line 150: So the conclusion from this section is that *O. cornifrons* shows a preference for Rosaceae spp generally (not a preference for E-A spp)?

- Yes mostly, hence we phrased it as we did, as they are not obligate to East Asian species. As we point out in the discussion, East-Asian Rosaceae seem to be the most "preferred", but we discuss the conditions that may alter that.

Line 165: Other rosaceae? Of remotely similar abundance to the apples?

- There are plenty of different species blooming in the rows of apple orchards and in the forest and edges by where they were placed; see Kammerer et al 2016 which we make reference to in the discussion. Please see our added paragraph in lines 203-222.

Line 170: I'm not sure what the authors mean here, given that most of the manuscript seems to be about (relative) abundance. Testing whether the metabarcoding was an accurate representation of the pollen provisions? That plant species abundance in the environment wasn't tested (which is what I at first thought)?

- Apologies for the wrong choice of words here, we meant that the plant species abundance in the field was not measured. We clarified this.

Line 172: They might indicate this, but this seems some distance from obtaining firmer proof.

- Agreed, but worth stating as part of the discussion

Line 175: Was this the case, or is it speculation?

- Speculation, we changed the sentence however for a revised paragraph in lines 217-222:
- "The use of non-Asian host plants may be the result of their sampling the landscape, a byproduct of nectar foraging as opposed to active pollen foraging, a dearth of Rosaceae and Fabaceae pollen post-crop bloom in orchards, or that alternative pollen resources are needed in their diet. How bees adjust their foraging behavior in absence of preferred resources, or the effects of local floral abundance on species preferences warrants further investigation in more controlled settings (see [22,44])."

Line 180: is it really phylogenetic? Is that the case they are making?

- We rephrased the sentence in lines 224-225 to say "The ability of mesolectic bee species to exclusively use host-plant species within family level preferences supports that these pollinators are ideal for crop pollination [17]."

Line 182: I think this sentence could be modified slightly to make it clear why an even pollen load is important (if it is).

- We modified the sentence in lines 225-228:
- "In apple orchards, the commercial apple (*Malus pumila*) and its East-Asian pollinizer crabapple (*M. baccata*) were represented evenly in pollen loads, indicating that *O. cornifrons* is effectively delivering pollination services [12], likely visiting both species and transferring pollen within foraging bouts throughout the day."

Line 185: correlation only?

- We made adjustments to this paragraph in lines 231-236:

- "However, we did not observe the same degree of preference for *Prunus* pollen in cherry orchards as we observed for *Malus* pollen in apple orchards (Figure 2). In cherry orchards, the bees also visited alternative rosaceous pollen sources (including *Malus*, *Rosa*, and *Amelanchier*) which may result from the short cherry blooming phenology or the higher preference for East-Asian pollen from Rosaceae plants."

Line 194: Really? Don't the results show that *O. cornifrons* will happily feed on non E-A spp if that is what is available?

- We made some modifications to this paragraph in lines 247-257:
- "The quantitative and qualitative results of pollen use that we report in this study suggest that *O. cornifrons* may not be directly competing for floral resources with other native *Osmia* spp. [45,46]. The Eastern North American native *O. lignaria lignaria* prefers early blooming trees such as *Quercus* and *Salix* [45]. Western *O. californica* exhibit preference for *Hydrophyllum*, and *O. l. propinqua* for *Hydrophyllum*, *Salix*, and the widespread *Crataegus* [48,49]. These preferred pollen hosts from North American *Osmia* spp. are absent from our dataset even though they bloom during *O. cornifrons* foraging period [28]. Potential competitive foraging between *O. cornifrons* and *O. l. lignaria* on *Cercis* spp. could occur as indicated by the preference for *Cercis* pollen in some of our samples [20,22,24,46,47]. However, specific and detailed studies would need to determine the level and effect of competition between species [15,17,46] (including competition for nesting sites in proximity to preferred hosts)."

Line 199: This sounds like better evidence. But were they available in the region and at the time (if not resources were not measured)?

- Yes, they were. We added some more info and reference to the sentence in lines 252-253: "These preferred pollen hosts from North American *Osmia* spp. are absent from our dataset even though they bloom during *O. cornifrons* foraging period [28]."

Line 201: But this hardly equates with the v high use of cercis (if RA is accurate).

Line 203: Is this relevant to pollen pref?

- We edited the conclusion of the paragraph to clarify in lines 253-257:
- "Potential competitive foraging between *O. cornifrons* and *O. l. lignaria* on *Cercis* spp. could occur as indicated by the preference for *Cercis* pollen in some of our samples [20,22,24,46,47]. However, specific and detailed studies would need to determine the level and effect of competition between species [15,17,46] (including competition for nesting sites in proximity to preferred hosts)."

Line 208: within the field, not outside?

- Correct, we clarified this

Line 224: At what phylogenetic depth are the authors referring to?

- Family level, which we added